# The immune factors driving DNA methylation variation in human blood

Jacob Bergstedt [1,2,3] ✉, Sadoune Ait Kaci Azzou [1], Kristin Tsuo [1], Anthony Jaquaniello[1], Alejandra Urrutia[4], Maxime Rotival [1], David T. S. Lin[5], Julia L. MacIsaac[5], Michael S. Kobor [5], Matthew L. Albert [4], Darragh Duffy [6], Etienne Patin [1,53] ✉, Lluís Quintana-Murci [1,7,53] ✉ & Milieu Intérieur Consortium*

Epigenetic changes are required for normal development, yet the nature and respective contribution of factors that drive epigenetic variation in humans remain to be fully characterized. Here, we assessed how the blood DNA methylome of 884 adults is affected by DNA sequence variation, age, sex and 139 factors relating to life habits and immunity. Furthermore, we investigated whether these effects are mediated or not by changes in cellular composition, measured by deep immunophenotyping. We show that DNA methylation differs substantially between naïve and memory T cells, supporting the need for adjustment on these cell-types. By doing so, we find that latent cytomegalovirus infection drives DNA methylation variation and provide further support that the increased dispersion of DNA methylation with aging is due to epigenetic drift. Finally, our results indicate that cellular composition and DNA sequence variation are the strongest predictors of DNA methylation, highlighting critical factors for medical epigenomics studies.

Epigenetic research has improved our understanding of the existing links between environmental risk factors, aging, genetic variation, and human disease[1,2]. Epigenome-wide association studies (EWAS) have shown that DNA methylation (i.e., 5-methylcytosine, 5mC), the most studied epigenetic mark in humans, is associated with a wide range of environmental exposures along the life course, such as chemicals[3] or past socioeconomic status[4–7]. Changes in DNA methylation have also been associated with non-communicable diseases, such as Parkinson's and Alzheimer's diseases, multiple sclerosis, systemic lupus erythematosus, type 2 diabetes and cardiovascular disease[8–11]. These studies collectively suggest that DNA methylation marks could be of tremendous value as gauges of the exposome and as clinical biomarkers[12,13].

However, interpretation of EWAS remains limited. First, because the epigenome of a cell reflects its identity[14,15], a risk factor or a disease that alters cellular composition also alters 5mC levels measured in the tissue[16]. It is thus necessary to determine if an exposure affects cellular composition or DNA methylation states of cell types, in order to better understand the link between such an exposure, DNA methylation and disease[17]. Previous studies have accounted for cellular heterogeneity in blood by using cell sorting experiments, or cellular proportions estimated from 5mC profiles through in-silico cell mixture deconvolution

[1]Institut Pasteur, Université Paris Cité, CNRS UMR2000, Human Evolutionary Genetics Unit, Paris, France. [2]Unit of Integrative Epidemiology, Institute of Environmental Medicine, Karolinska Institutet, Stockholm, Sweden. [3]Department of Medical Epidemiology and Biostatistics, Karolinska Institutet, Stockholm, Sweden. [4]HI-Bio, South San Francisco, CA, USA. [5]Edwin S.H. Leong Healthy Aging Program, Centre for Molecular Medicine and Therapeutics, Department of Medical Genetics, University of British Columbia, Vancouver, Canada. [6]Institut Pasteur, Université Paris Cité, Translational Immunology Unit, Institut Pasteur, Paris, France. [7]Chair of Human Genomics and Evolution, Collège de France, Paris, France. [53]These authors contributed equally: Etienne Patin, Lluís Quintana-Murci. *A list of authors and their affiliations appears at the end of the paper. ✉e-mail: jacob.bergstedt@ki.se; epatin@pasteur.fr; quintana@pasteur.fr

techniques[18,19], but these approaches focus on a subset of frequent cell types that capture only a part of blood cellular composition. Second, the strong links between DNA methylation and DNA sequence variation, attested by the numerous DNA methylation quantitative trait loci (meQTLs) detected so far[20–23], suggest that environmental effects on the epigenome may operate through gene-by-environment interactions, but evidence for such interactions remains circumstantial. Finally, environmental risk factors with a yet-unknown effect on DNA methylation, such as common infections, could confound associations between other risk factors, DNA methylation and human phenotypes. Thus, a detailed study of the factors that impact DNA methylation at the population level, and the extent to which their effects are mediated by changes in cellular composition, is required to understand the role of epigenetic variation in health and disease.

To address this gap, we generated whole blood-derived DNA methylation profiles at >850,000 CpG sites for 884 healthy adults of the Milieu Intérieur cohort. We leveraged the deep characterization of the cohort, including high-resolution immunophenotyping by flow cytometry[24,25], to determine whether and how cellular composition, intrinsic factors (i.e., age and sex), genetic variation, and 139 health- and immunity-related variables and environmental exposures affect the blood DNA methylome. We first assessed differences in the DNA methylation profiles of 16 different immune cell types. We then performed EWAS, adjusted or not for the measured proportions of the 16 immune cell subsets, and mediation analyses to robustly delineate effects on DNA methylation that are direct, i.e., acting through changes within cells, from those that are mediated, i.e., acting through subtle changes in cellular composition[26]. We show that adjusting EWAS for 16 measured cell proportions better accounts for cellular heterogeneity than current cell mixture deconvolution methods. We identify latent cytomegalovirus (CMV) infection as a key factor affecting population variation in 5mC levels, through the dysregulation of human transcription factors and profound changes in the proportion of differentiated T cells. We show that the increased dispersion of DNA methylation with aging is independent of cellular composition, supporting instead a decrease in the fidelity of the epigenetic maintenance machinery. Furthermore, we show that a large part of the effects on DNA methylation of aging, smoking, CMV serostatus, and chronic low-grade inflammation is due to subtle changes in blood cell composition, and characterize the DNA methylation signature of cell types affected by these factors. Finally, we find that the largest effects on DNA methylation are due to DNA sequence variation, whereas the most widespread differences among individuals are the result of blood cellular heterogeneity. This work generates new hypotheses about mechanisms underlying DNA methylation variation in the human population and highlights critical factors to be considered in medical epigenomics studies.

## Results

### Proportions of naïve and differentiated T cells markedly contribute to DNA methylation variation

To investigate the non-genetic and genetic factors that affect population variation in DNA methylation, we quantified 5mC levels at >850,000 CpG sites, with the Illumina Infinium MethylationEPIC array, in the 1000 healthy donors of the Milieu Intérieur cohort (Fig. 1a). The cohort includes individuals of Western European origin, equally stratified by sex (i.e., 500 women and 500 men) and age (i.e., 200 individuals from each decade between 20 and 70 years of age), who were surveyed for detailed demographic and health-related information[24], including factors that are known to affect DNA methylation (i.e., age, sex, smoking, BMI and socioeconomic status), that have been proposed to affect DNA methylation (e.g., dietary habits, upbringing) or that pertain to the immune system (e.g., past and latent infections, past vaccinations, antibody levels; Supplementary Data 1). All donors were genotyped at 945,213 single-nucleotide polymorphisms (SNPs),

yielding 5,699,237 accurate SNPs after imputation[25]. After quality control filtering, high-quality measurements of DNA methylation were obtained at 644,517 CpG sites for 884 unrelated individuals[27] (Supplementary Fig. 1; Methods). We found that 5mC levels well reproduce expected patterns across chromatin states[15], supporting the good quality of the data (Supplementary Fig. 1 and Supplementary Notes).

Whereas most epigenome-wide studies adjust on estimated cellular composition to detect direct effects on DNA methylation (i.e., acting through changes within cells), we sought to assess both direct effects and effects that are mediated by changes in cellular composition, as the genomic location and magnitude of mediated effects can inform us about how cell differentiation is regulated in response to environmental exposures[17]. We thus measured, in all donors, the proportions of 16 immune cell subsets by standardized flow cytometry, including neutrophils, basophils, eosinophils, monocytes, natural killer (NK) cells, dendritic cells, B cells, CD4⁻CD8⁻ T cells and naive, central memory (CM), effector memory (EM) and terminally differentiated effector memory cells (EMRA) CD4⁺ and CD8⁺ T cells[25].

We first determined which immune cell populations most affect DNA methylation variation, by quantifying differences in 5mC levels between the 16 blood cell subsets with multivariable regression models including log-ratios of cell subsets, defined according to the hierarchical and compositional nature of the data[28] (Methods). We verified that our models are accurate, using simulations and comparisons with independent DNA methylation data from sorted cellular subsets[29]. We found that our estimated effects of cell subset log-ratios on 5mC levels perform as expected on simulated data (Supplementary Fig. 2 and Supplementary Notes) and are highly correlated with DNA methylation differences observed between sorted immune cell fractions ($R > 0.6$; Supplementary Data 2). When applying these models on our data, we found that 5mC levels of 134,079 CpG sites (20.8% of CpG sites, Supplementary Data 2) are associated with the log-ratio of myeloid vs. lymphoid lineages (Bonferroni corrected $P_{adj} < 0.05$). Furthermore, the log-ratio of these subsets is the factor most associated with the first three Principal Components (PCs) of the DNA methylation data (multiple linear mixed model of PC1: $P = 5.0 \times 10^{-18}$; PC2: $P = 1.6 \times 10^{-43}$; PC3: $P = 6.7 \times 10^{-17}$), which respectively explain 11.4, 7.5, and 5.5% of variation in DNA methylation. Importantly, we also found that 20,758 and 44,919 CpG sites are associated with the log-ratios of naïve and differentiated (CM, EM and EMRA) CD4⁺ and CD8⁺ T cell subsets, respectively ($P_{adj} < 0.05$, Supplementary Data 2), supporting the view that 5mC levels differ substantially among T cell subpopulations[30,31]. Furthermore, the log-ratios of naïve and differentiated CD4⁺ and CD8⁺ subsets are also associated with PC1 and PC3 ($P < 1.2 \times 10^{-4}$; Fig. 1c, d). These results indicate that differences in the proportion of naïve and differentiated subsets of CD4⁺ and CD8⁺ T cells contribute substantially to DNA methylation variation and may mediate associations between DNA methylation and environmental exposures or diseases.

### Cell mixture deconvolution methods partially account for blood cell heterogeneity

Direct effects of environmental exposures or diseases on DNA methylation are often estimated by adjusting EWAS on major cell-type fractions, which are predicted in silico from 5mC levels with cell mixture deconvolution methods[18,32]. However, standard methods only predict the overall proportions of CD4⁺ and CD8⁺ T cells and may therefore overestimate the direct effects on DNA methylation of factors that affect T cell composition, such as aging and viral infections[25,33]. To test this hypothesis, and to assess more generally how intrinsic and environmental factors affect the DNA methylome, we conducted EWAS of 141 candidate factors, by using linear mixed models adjusted on batch variables, genetic factors (i.e., associated meQTL variants), genetic ancestry, smoking status,

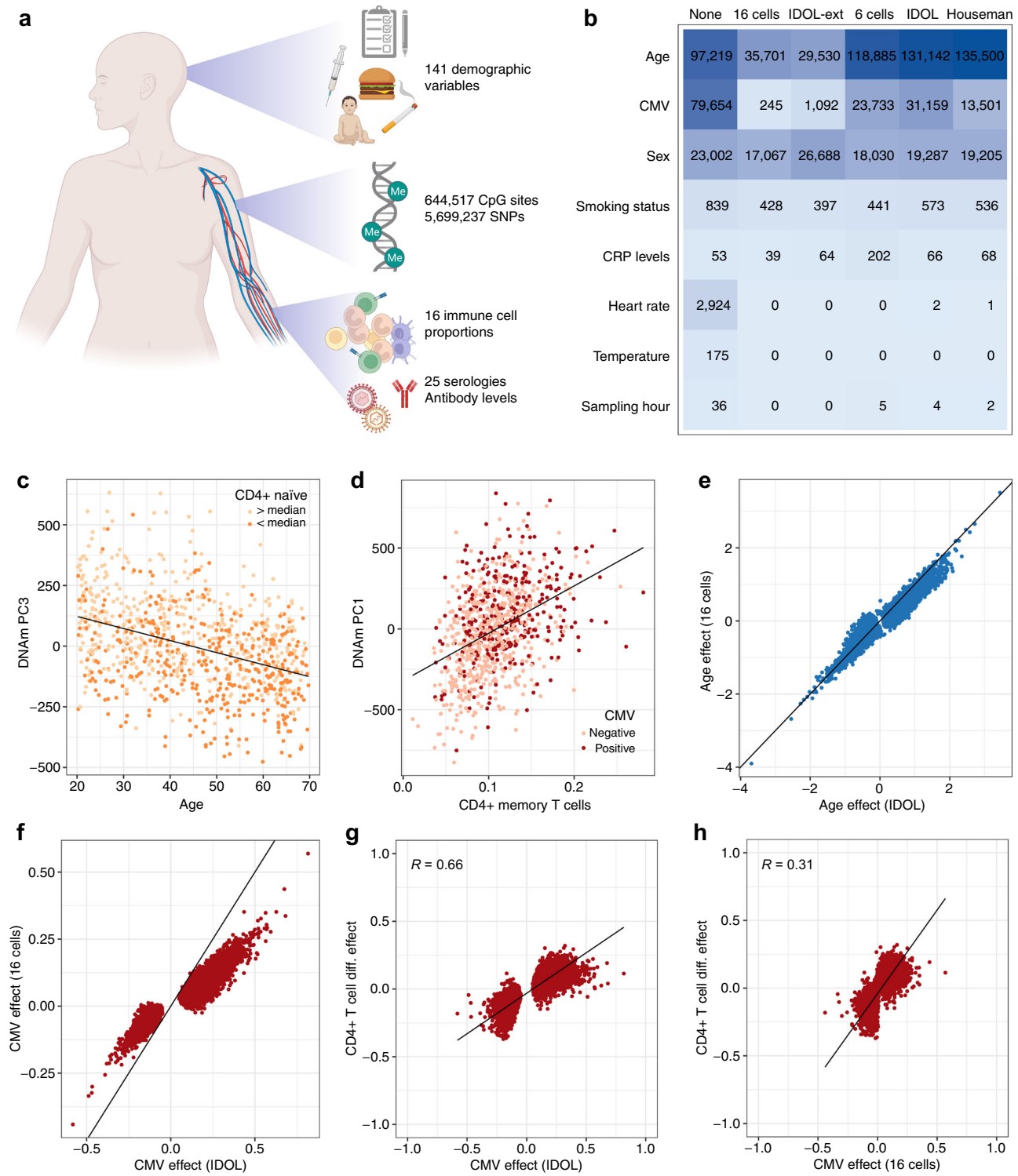

sex and a non-linear age term (Methods). Models were adjusted, or not, for the 16 measured cell proportions, to estimate total (i.e., direct and mediated) or direct effects, respectively. Mediated effects were estimated by mediation analysis[34] (Methods). We considered that each EWAS constitutes a separate family of association tests and used the Bonferroni correction for multiple testing adjustment ($P_{adj} < 0.05$).

Out of the 141 candidate factors, those that have significant total effects on DNA methylation include age ($n = 97,219$ CpG sites; 15.1% of CpG sites), cytomegalovirus (CMV) serostatus ($n = 79,654$; 12.4%), sex ($n = 23,002$; 3.6%), heart rate ($n = 2,924$; 0.5%), smoking ($n = 839$; 0.1%),

body temperature ($n = 175$), C-reactive protein (CRP) levels ($n = 53$), the hour of blood draw ($n = 36$) and traits related to lipid metabolism ($n = 3$; Fig. 1b and Supplementary Data 1). Accordingly, the first PCs of DNA methylation are most strongly associated with CMV (PC1: $P = 8.3 \times 10^{-13}$; PC2: $P = 7.8 \times 10^{-10}$), age (PC3: $P = 5.7 \times 10^{-29}$) and sex (PC4: $P = 2.2 \times 10^{-5}$), when not considering immune cell fractions (Fig. 1c, d and Supplementary Fig. 1i, j). When adjusting on blood cell composition, factors that have significant direct effects on DNA methylation include age ($n = 35,701$; 5.5%), sex ($n = 17,067$; 2.6%), smoking ($n = 428$; 0.07%), CMV serostatus ($n = 245$; 0.04%), CRP levels ($n = 39$) and lipid metabolism-related traits ($n = 3$; Fig. 1b,

**Fig. 1 | Non-genetic effects on the blood DNA methylome according to different corrections for cellular heterogeneity. a** Study design. Created with BioRender.com. **b** Number of CpG sites associated with non-genetic factors, according to different corrections for cellular heterogeneity. Columns indicate adjustments for 16 blood cell proportions measured by flow cytometry ("16 cells"), 12 blood cell proportions estimated by the EPIC IDOL-Ext deconvolution method[29] ("IDOL-ext"), 6 blood cell proportions measured by flow cytometry ("6 cells"), 6 cell proportions estimated by the IDOL deconvolution method[32] ("IDOL"), 6 cell proportions estimated by Houseman et al.'s deconvolution method[18] ("Houseman") and no adjustment for blood cell composition ("None"). Tests were corrected for multiple testing by the Bonferroni adjustment. **c** Age against the third Principal Component (PC) of DNA methylation levels. Colors indicate donors whose proportion of naïve CD8[+] T cells in blood is below or above the cohort median. **d** Proportion of CD4[+] memory T cells against the first PC of DNA methylation levels. Colors indicate the CMV serostatus of donors. **e** Direct effects of age on 5mC levels, adjusting on 6 cell proportions estimated by IDOL, against direct effects of age on 5mC levels, adjusting on 16 cell proportions measured by flow cytometry. **f** Direct effects of CMV serostatus on 5mC levels, adjusting on 6 cell proportions estimated by IDOL, against direct effects of CMV serostatus on 5mC levels, adjusting on 16 cell proportions measured by flow cytometry. **g** Effects of CD4[+] T cell differentiation on 5mC levels against direct effects of CMV serostatus on 5mC levels, adjusting on 6 cell proportions estimated by IDOL. **h** Effects of CD4[+] T cell differentiation on 5mC levels against direct effects of CMV serostatus on 5mC levels, adjusting on 16 cell proportions measured by flow cytometry. **e–h** Effect sizes are given in the M value scale. Only associations significant either with the model adjusting for IDOL-estimated cell proportions or the model adjusted for 16 measured cell proportions are shown ($P_{adj} < 0.05$). **e, f** The black line indicates the identity line. **c, d, g, h** The black line indicates the linear regression line. Statistics were computed based on a sample size of $n = 884$ and for 644,517 CpG sites.

Supplementary Fig. 3 and Supplementary Notes). These results suggest that, whereas most CMV effects are mediated by cellular composition, the effects of sex on DNA methylation are mainly direct, and a substantial direct effect of age is also retained, even after adjusting for naïve and memory CD4[+] and CD8[+] T cell subsets. Accordingly, first PCs of DNA methylation remain associated with sex (PC4: $P = 1.3 \times 10^{-3}$) and age (PC3: $P = 1.1 \times 10^{-9}$; Fig. 1c), when considering immune cell fractions, but not with CMV serostatus (PC1: $P > 0.05$; Fig. 1d). No significant direct effects of heart rate, body temperature and hour of sampling were detected, indicating that the effects of these factors on DNA methylation are due exclusively to changes in immune cell composition[35,36].

We then evaluated the performance of three reference-based in silico cell mixture deconvolution methods: Houseman et al.'s method, IDOL, and EPIC IDOL-Ext[18,29,32]. We observed that cell proportions estimated by the three methods are substantially correlated with measured cell proportions (Supplementary Fig. 4). We then compared EWAS results adjusted either on our flow cytometric data or on cell proportions estimated by the three deconvolution methods. We found that EWAS adjusted by the IDOL method detects more CpG sites associated with most candidate factors, relative to EWAS adjusted on the measured proportions of 16 cell types, particularly for age ($n = 131,142$ vs. 35,701) and latent CMV infection ($n = 31,159$ vs. 245) (Fig. 1b, e, f). Similar results were found with Houseman's method (Fig. 1b). Accordingly, the first PC of DNA methylation remains strongly associated with CMV serostatus and age when adjusting on IDOL cellular fractions ($P = 7.5 \times 10^{-6}$ and $P = 3.2 \times 10^{-17}$, respectively), whereas it is not when considering 16 measured cell proportions ($P > 0.01$). Conversely, EWAS adjusted by the EPIC IDOL-Ext method, which estimates subsets of naïve and memory CD4[+] and CD8[+] T cell populations[29], provide results that are similar to those of EWAS adjusted for high-resolution flow cytometric data (Fig. 1b). These results suggest that first-generation deconvolution methods do not fully distinguish direct effects on DNA methylation from those that are mediated by fine-grained changes in blood cell composition.

To further test this scenario, we conducted EWAS adjusted on flow cytometric data for only six major cell types and found results comparable to those for Houseman et al.'s and the IDOL methods (Fig. 1b). Furthermore, CMV effect sizes adjusted on IDOL cellular fractions or the six major cell proportions were twice more correlated with estimated measures of DNA methylation differences between naïve and differentiated CD4[+] T cells, relative to CMV effect sizes adjusted on 16 measured cell proportions ($R = 0.66$, relative to $R = 0.31$, respectively; Fig. 1g, h). Together, these results indicate that adjustment for the proportions of only the six major cell types is not able to fully account for blood cell heterogeneity, particularly when estimating the effects of age and CMV infection on DNA methylation, two factors that are known to skew CD4[+] and CD8[+] T cell compartments toward differentiated phenotypes[25].

## Cytomegalovirus infection alters the blood DNA methylome through the regulation of host transcription factors

We identified CMV serostatus as one of the exposures that is associated with the largest number of CpG sites (Fig. 1b). CMV is the causative agent of a latent, mainly asymptomatic, infection that ranges in seroprevalence from 30 to 100% across populations[37]. CMV is known to drastically alter the composition of the CD4[+] and CD8[+] T cell compartments in blood[25,33]. Accordingly, we found that 85,922 CpG sites show a significant cell-composition-mediated effect of CMV serostatus on DNA methylation ($P_{adj} < 0.05$; Supplementary Data 1), indicating that the effects of the latent infection are mainly mediated by cellular composition. Furthermore, we observed a strong correlation between mediated and total effect sizes of CMV serostatus ($R = 0.93$; Fig. 2a) and 99.5% of CpG sites with a significant direct effect also show a significant mediated effect ($n = 244/245$). We found that mediated effect sizes of CMV are strongly correlated with estimated measures of DNA methylation differences between naïve and memory CD4[+] and CD8[+] T cells ($R = 0.68$ and $R = 0.53$, respectively; Fig. 2b), suggesting that cell-composition-mediated effects of CMV are predominantly attributable to changes in these T cell subsets.

One of the strongest cell-composition-mediated effects of CMV infection was observed in an intron of *DNMT3A* (β value scale 95% confidence interval [CI]: [1.8%, 2.4%], $P_{adj} = 1.1 \times 10^{-23}$), encoding a key DNA methyltransferase playing a role in the replication of some herpesviruses[38]. CMV[+] donors show a substantial increase in the proportion of CD4[+] and CD8[+] $T_{EMRA}$ cells ($P = 6.8 \times 10^{-35}$ and $P = 1.9 \times 10^{-50}$, respectively), which in turn are associated with higher 5mC levels at *DNMT3A* ($P = 3.3 \times 10^{-25}$ and $P = 1 \times 10^{-53}$, respectively), supporting mediation by differentiated memory T cell subsets (Fig. 2c). To test if the effects of CMV infection on 5mC levels are cell-type-dependent, we derived and verified an interaction model similar to CellDMC[39] (Methods). We restricted this analysis to interactions with the proportion of cells from the myeloid lineage, as previously reported[40], and found only one CpG site where CMV effects depend on the proportion of myeloid cells ($P_{adj} < 0.05$; Supplementary Data 3). These results indicate that CMV infection affects a large fraction of the blood DNA methylome primarily through changes in blood cell proportions, rather than through cell-type-dependent changes.

However, when adjusting for blood cell composition, including CD4[+] and CD8[+] T cell sub-types, a significant direct effect of CMV serostatus was detected for 245 CpG sites. Increased 5mC levels in CMV[+] donors localize predominantly in enhancers and regions flanking transcription start sites (odds ratio [OR] > 3.0, $P_{adj} < 5.3 \times 10^{-8}$; Supplementary Fig. 5), suggesting dysregulation of host gene expression as a result of latent infection. The second strongest direct effect of CMV infection was observed nearby the TSS of *LTBP3* (β value scale 95% CI: [1.9%, 3.1%], $P_{adj} = 7.1 \times 10^{-17}$; Fig. 2d and Supplementary Fig. 6). LTBP3 is a regulator of transforming growth factor β (TGF-β)[41], which is induced in CMV latently infected cells[42]. Strikingly, CpG sites showing

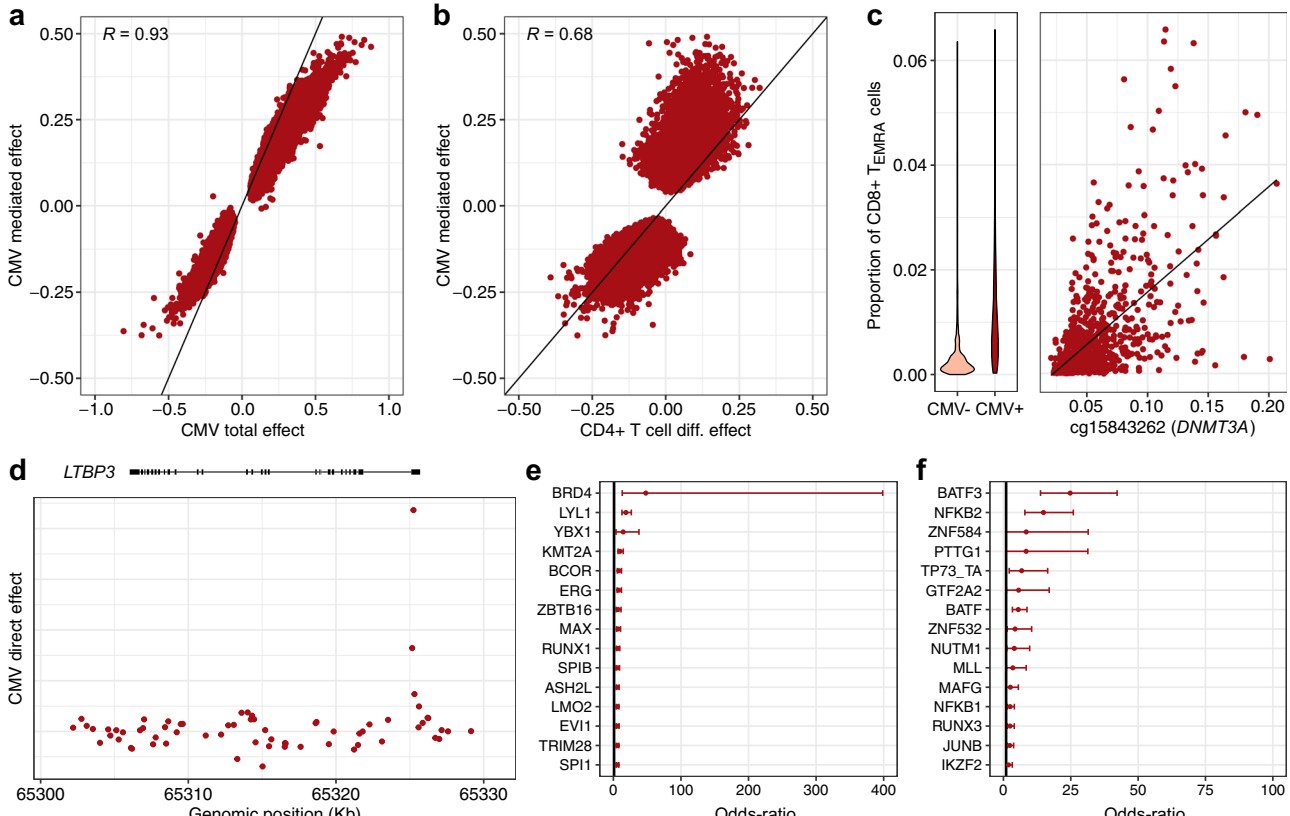

**Fig. 2 | Effects of cytomegalovirus infection on the blood DNA methylome.**
**a** Total effects against cell-composition-mediated effects of CMV infection on 5mC levels. **b** Effects of CD4+ T cell differentiation on 5mC levels against cell-composition-mediated effects of CMV infection on 5mC levels. **c** Proportion of CD8+ $T_{EMRA}$ cells in CMV− and CMV+ donors (left panel). 5mC levels at the *DNMT3A* locus against the proportion of CD8+ $T_{EMRA}$ cells (right panel). 5mC levels are given in the β value scale. The black line indicates the linear regression line. **d** Genomic distribution of direct effects of CMV infection at the *LTBP3* locus. **e** Enrichment of CpG sites with a significant direct, positive effect of CMV infection in binding sites for TFs. **f** Enrichment of CpG sites with a significant direct, negative effect of CMV infection in binding sites for TFs. **a**, **b** Only CpG sites with a significant cell-composition-mediated effect are shown. The black line indicates the identity line. Tests were corrected for multiple testing by the Bonferroni adjustment. **a**, **b**, **d** Effect sizes are given in the M value scale. **e**, **f** The 15 most enriched TFs are shown, out of 1165 tested TFs. The point and error bars indicate the odds-ratio and 95% CI. CIs were estimated by the Fisher's exact method. Statistics were computed based on a sample size of *n* = 884 and for 644,517 CpG sites.

increased 5mC levels in CMV+ donors are strongly enriched in binding sites for the BRD4 transcription factor (TF) (*n* = 187/189, OR = 48.0, 95% CI: [13.1, 399.0], $P_{adj} < 1.1 \times 10^{-27}$; Fig. 2e and Supplementary Data 4), a bromodomain protein that plays a critical role in the regulation of latent and lytic phases of CMV infection[43]. Conversely, CpG sites showing a decrease in DNA methylation in CMV+ donors are strongly enriched in binding sites for BATF3 (OR = 24.8, 95% CI: [13.8, 42.2], $P_{adj} < 1.3 \times 10^{-14}$; Fig. 2f), which is paramount in the priming of CMV-specific CD8+ T cells by cross-presenting dendritic cells[44]. Collectively, these analyses imply that CMV infection directly affects the human blood DNA methylome through the dysregulation of host TFs implicated in viral latency and host immune response.

Finally, to motivate future research on the epigenetic effects of CMV infection, we used elastic net regression and stability selection to predict CMV serostatus from DNA methylation (Methods). Based on 547 CpG sites, the model predicts CMV serostatus with an out-of-sample accuracy of 87%, using 10-fold cross-validation. We anticipate that this model will be useful to determine if latent CMV infection can confound epigenetic risk for disease[45,46].

**Aging elicits DNA hypermethylation related to Polycomb repressive complexes and increased epigenetic dispersion**
Although the effects of aging on DNA methylation are well established[47–51]; it remains unclear the extent to which they are due to changes in unmeasured proportions of differentiated T cells (Fig. 1b)

or CMV infection, which are both strongly associated with age[25,52]. Indeed, age has a significant total effect on 5mC levels at 97,219 and 113,742 CpG sites, when adjusting or not on CMV serostatus, and CMV infection mediates a substantial fraction of total age effects (*n* = 10,074 CpG sites). We thus investigated how the blood DNA methylome is shaped by the intertwined processes of cellular aging (i.e., direct effects) and age-related changes in blood cellular composition (i.e., mediated effects), while accounting for CMV serostatus.

We found that, out of the 35,701 CpG sites associated directly with age, more than 97% were associated with age in a previous EWAS[53], indicating a strong overlap (OR 95% CI: [35.6, 40.8]). In line with previous findings[54], direct effects of age are typically larger than mediated effects (Fig. 3a). Furthermore, the strongest direct age effects, such as those observed at *ELOVL2* and *FHL2* (Supplementary Fig. 6), are not mediated by cellular composition ($P_{adj}$ = 1.0), suggesting that age-related changes at these CpG sites are typically shared across cell-types. We observed that 61% of the CpG sites directly associated with age show a decrease in 5mC levels. Age-associated demethylation predominates outside of CpG islands (CGIs) and in regions flanking transcription start sites and in enhancers (Fig. 3b and Supplementary Fig. 7a, b). Conversely, DNA hypermethylation was observed in 95% of age-associated CpGs within CGIs. Consistently, CpG sites exhibiting increasing 5mC levels with age are mainly found in Polycomb-repressed regions, bivalent TSSs, and bivalent enhancers (Fig. 3b, c), which are CGI-rich

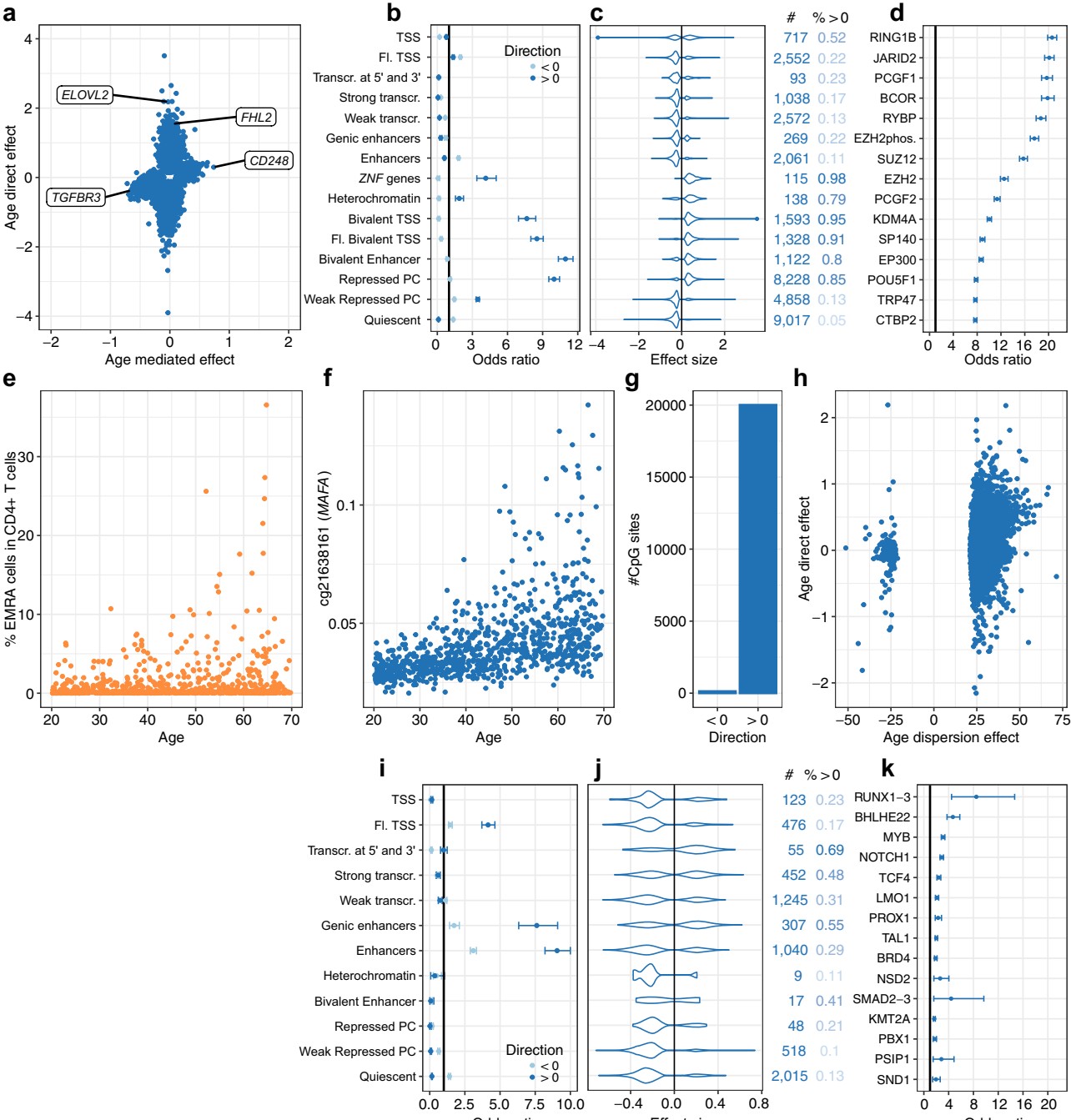

**Fig. 3 | Direct and cell-composition-mediated effects of aging on the blood DNA methylome. a** Direct effects against cell-composition-mediated effects of age on 5mC levels (50-year effect). Only CpG sites with a significant direct or cell-composition-mediated effect are shown. Labels denote genes with strong direct or cell-composition-mediated effects of age. **b** Enrichment in CpG sites with significant direct effects of age, across 15 chromatin states. **c** Distributions of significant direct effects of age, across 15 chromatin states. **d** Enrichment of CpG sites with a significant positive, direct effect of age in binding sites for TFs. **e** Increased variance of the proportion of CD4⁺ T_EMRA cells with age. **f** Increased variance of 5mC levels with age at the *MAFA* locus. 5mC levels are given in the β value scale. **g** Number of CpG sites with a significant increase or decrease in variance with age. **h** Direct effects against dispersion effects of age on 5mC levels. **i** Enrichment of CpG sites with significant cell-composition-mediated effects of age, across 12 chromatin states. **j** Distributions of significant cell-composition-mediated effects of age, across 12 chromatin states. **k** Enrichment of CpG sites with significant cell-composition-mediated, positive effects of age in binding sites for TFs. **a**, **c**, **h**, **j** Effect sizes are given in the M value scale. **c**,**j** Numbers on the right indicate the number of associated CpG sites and proportion of positive effects. **b**, **d**, **i**, **k** The point and error bars indicate the odds-ratio and 95% CI. CIs were estimated by the Fisher's exact method. Statistics were computed based on a sample size of $n = 884$ and for 644,517 CpG sites. **d**, **k** The 15 most enriched TFs are shown, out of 1165 tested TFs. **b**, **c**, **i**, **j** Chromatin states were defined in PBMCs[15]. Chromatin states were not shown when <5 associated CpG sites were observed. TSS, Fl. and PC denote transcription start site, flanking and Polycomb, respectively. Statistics were computed based on a sample size of $n = 884$ and for 644,517 CpG sites. Tests were corrected for multiple testing by the Bonferroni adjustment.

regions (Supplementary Fig. 1M, N). Furthermore, these CpG sites are most enriched in binding sites for RING1B, JARID2, RYBP, PCGF1, PCGF2, and SUZ12 TFs (OR > 10.0; Fig. 3d and Supplementary Data 4), which are all part of the Polycomb repressive complexes 1 and 2. PRC1 and PRC2 mediate cellular senescence and modulate longevity in invertebrates[55,56]. Importantly, when restricting the analysis to CpG sites outside of CpG islands, we found similar enrichments in Polycomb-repressed regions (OR 95% CI [17.7, 20.0]) and PRC TF binding sites (RING1B OR 95% CI: [19.9, 22.4]; PCGF2 OR 95% CI [17.8, 20.7]). Finally, genes with age-increasing 5mC levels are strongly enriched in developmental genes ($P_{adj} = 1.7 \times 10^{-48}$; Supplementary Data 5), which are regulated by PRCs[57]. Overall, these results confirm previously described effects of age on the blood DNA methylome, while accounting more comprehensively for blood cell composition and CMV infection, and support a key regulatory role of Polycomb proteins in age-related hypermethylation[58].

We then assessed whether age-related changes in blood cell composition or CMV seropositivity could contribute to age-related changes in the variance of 5mC levels, a phenomenon known as "epigenetic drift" (i.e., the divergence of the DNA methylome as a function of age owing to stochastic changes)[51,59–61]. We observed that the proportion of several cell types in blood is increasingly dispersed with aging, such as $CD4^+$ $T_{EMRA}$ cells (Fig. 3e). Therefore, we fitted models parameterizing the residual variance with a linear age term, and adjusting for 16 immune cell proportions, age, CMV serostatus, smoking status and sex in the mean function (Methods). We observed a significant dispersion of DNA methylation with age for 3.1% of all CpG sites ($n = 20,140$, $P_{adj} < 0.05$). We compared these CpG sites with those previously reported to be increasingly variable with age in whole blood and monocytes[60] and replicated 2604 out of 5,075 CpG sites, supporting a strong overlap between the two different approaches (OR 95% CI: [36.2, 40.8]). An example of a CpG site with a large, age-increasing dispersion is found in the TSS of *MAFA* ($P_{adj} = 4.4 \times 10^{-43}$; Fig. 3f), encoding a transcription factor that regulates insulin. Strikingly, 99.4% of CpGs with age-related dispersion show an increase in the variance of 5mC levels with age (Fig. 3g), supporting a decrease in the fidelity of epigenetic maintenance associated with aging. In addition, we found that, out of 20,140 CpG sites with age-related dispersion, 87.3% show no significant changes in mean 5mC levels with age, and we detected no correlation between estimates of dispersion and direct age effect sizes (Fig. 3h), implying that these results are not driven by relationships between the average and variance of 5mC levels. Furthermore, when also adjusting the variance function for cellular composition, we found evidence of dispersion in 8,576 CpG sites ($P_{adj} < 0.05$), with similar effect sizes as in the previous model ($R = 0.93$; Methods). Collectively, these findings indicate that aging elicits numerous DNA methylation changes in a cell-composition-independent manner, including global epigenome-wide demethylation, hypermethylation of PRC-associated regions and increased variance, highlighting the occurrence of different mechanisms involved in epigenetic aging.

### Immunosenescence-related changes in cellular composition mediate DNA methylation variation with age

We detected a significant cell-composition-mediated effect of age at ~1.1% of CpG sites ($n = 7090$; Fig. 3a and Supplementary Data 1), indicating that a substantial fraction of age-related changes in DNA methylation are due to age-related changes in immune cell proportions. Mediated effects are most often associated with demethylation (76% of age-associated CpG sites), regardless of the chromatin state or CGI density of the loci considered (Fig. 3j and Supplementary Fig. 7c, d). Enhancers and regions flanking transcription start sites are enriched in CpG sites with a significant cell-composition-mediated effect

of age (Fig. 3i), possibly because these regions tend to be regulated in a cell-type-dependent manner[15]. In contrast with direct age effects, CpG sites with a cell-composition-mediated increase in DNA methylation are enriched in TF binding sites for RUNX1-3 (OR = 8.5, 95% CI: [4.5, 14.7], $P_{adj} < 1.2 \times 10^{-8}$), which are key regulators of hematopoiesis (Fig. 2k and Supplementary Data 4). Genes with CpG sites showing a mediated increase or decrease in DNA methylation with age are enriched in genes involved in lymphoid ($P_{adj} = 2.0 \times 10^{-7}$) and myeloid ($P_{adj} = 6.1 \times 10^{-13}$) cell activation, respectively (Supplementary Data 5). This indicates that mediated effects of age on DNA methylation are related to progressive, lifelong differences in the composition of the lymphoid and myeloid cell lineages.

We then determined if age effects on 5mC levels depend on the proportion of cells from the myeloid lineage, by using an interaction model (Methods). In line with a previous study[54], we found that cell-type-dependent effects of age (Supplementary Data 3) are limited; only 10 CpG sites show DNA methylation changes with age that depend on the proportion of myeloid cells ($P_{adj} < 0.05$; Supplementary Data 3). Importantly, age also has a strong mediated effect on all these CpG sites ($P_{adj} < 1.0 \times 10^{-10}$), implying that these loci are associated with age because of changes in blood cell composition, although their relation to age is cell-type-dependent. Collectively, our findings provide statistical evidence that DNA methylation variation with age results from different, non-mutually exclusive mechanisms: the progressive decline of the epigenetic maintenance system that is common to all cell types, the increased heterogeneity of immune cell subsets that characterizes immunosenescence[62] and, to a lesser extent, accelerated changes within specific blood cell compartments.

### Sex differences in DNA methylation are predominantly cell- and age-independent

Given that substantial differences in immune cell composition have been observed between women and men[25], we next assessed how cellular heterogeneity contributes to sex differences in DNA methylation[63–65]. We found 3.6% of CpG sites ($n = 23,002$) with a significant total effect of sex, 2.6% ($n = 17,067$) with a significant direct effect, and only 0.2% ($n = 1385$) with a significant cell-composition-mediated effect ($P_{adj} < 0.05$; Supplementary Fig. 8a and Supplementary Data 1). Out of CpG sites directly associated with sex, 96.2% were already associated with sex in a previous EWAS[53], indicating again a strong overlap (OR 95% CI: [39.6, 46.5]). The largest direct effects of sex were observed at *DYRK2*, *DNM1*, *RFTN1*, *HYDIN*, and *NAB1* genes ($P_{adj} < 1.0 \times 10^{-263}$; Supplementary Fig. 6). For example, the *DYRK2* promoter is 11.7% and 45.6% methylated in men and women, respectively, at a CpG site that we found to be bound by the X-linked PHF8 histone demethylase (Supplementary Fig. 8b, c). DYRK2 phosphorylates amino acids and plays a key role in breast and ovarian cancer development[66].

DNA methylation levels are higher in women at 79.7% of sex-associated autosomal CpG sites (Supplementary Fig. 8d, e), a pattern also observed in newborns[64]. This proportion is similar across different genomic regions, based on either chromatin states or CpG density (Supplementary Fig. 8e, g). When quantifying how sex differences in DNA methylation vary during adulthood, by adding a sex-by-age interaction term to our models (Methods), we found only 7 CpG sites with a significant, sex-dependent effect of age ($P_{adj} < 0.05$; Supplementary Data 3). Confirming previous findings[53,67], the strongest sex-by-age interaction effects were found at *FIGN* ($P_{adj} < 7.1 \times 10^{-15}$), associated with risk-taking behaviors[68] and educational attainment[69], and *PRR4* ($P_{adj} < 5.6 \times 10^{-3}$), associated with the dry eye syndrome, a hormone-dependent, late-onset disorder[70]. Overall, our findings indicate that the blood DNA methylome is widely affected by sex, but its effects are typically not mediated by cellular composition and do not change during adulthood.

## Gene × cell type and gene × environment interactions affect DNA methylation variation

Gene × environment interactions are thought to underlie adaptable human responses to environmental exposures through epigenetic changes[71]. To test if gene × environment interactions affect DNA methylation, we first estimated, for each CpG site, the effects on 5mC levels of local and remote DNA sequence variation, defined as genetic variants within a 100-Kb window and outside a 1-Mb window centered on the CpG site, respectively (Methods). We considered local and remote meQTLs to be independent families of tests and used the Bonferroni correction to adjust for multiple testing. We found a significant local meQTL for 107,048 CpG sites and a significant remote meQTL for 1228 CpG sites ($P_{adj} < 0.05$; Supplementary Fig. 9 and Supplementary Data 6). In agreement with previous studies[21,23], CpG sites with a local meQTL are enriched in enhancers (OR 95% CI: [2.09, 2.21]) and depleted in TSS and actively transcribed genes (OR 95% CIs: [0.52, 0.56] and [0.57, 0.60]; Fig. 4a). Conversely, CpG sites under remote genetic control are enriched in TSS regions (OR 95% CI: [2.10, 3.11]) and regions associated with *ZNF* genes (OR 95% CI: [1.26, 6.17]; Fig. 4b). Furthermore, we found that remote meQTL variants are also strongly concentrated in *ZNF* genes (OR 95% CI: [14.6, 29.8]; Fig. 4c), suggesting that zinc-finger proteins (ZFPs) play a role in the long-range control of DNA methylation, in line with their role in the regulation of heterochromatin[72–74].

We next explored whether effects of genetic variants on 5mC levels depend on the circulating proportion of myeloid cells. We found evidence for cell-type-dependent meQTLs at only 249 CpG sites ($P_{adj} < 0.05$; Fig. 4d and Supplementary Data 3), supporting the notion that genetic effects on 5mC levels are generally shared across blood cell subsets[75]. The strongest signal was found between 5mC levels upstream of *CLEC4C* and the nearby rs11055602 variant, which has been previously shown to strongly affect CLEC4C protein levels[76]. This C-type lectin, known as CD303, is used as a differentiation marker for dendritic cells, suggesting the epigenetic regulation of the locus is cell-type-dependent. Accordingly, rs11055602 genotype effects on DNA methylation depend on the circulating proportions of myeloid cells (β scale interaction effect, 95% CI: [0.16, 0.22], $P_{adj} = 7.4 \times 10^{-20}$; Fig. 4e), and dendritic cells (95% CI: CI: [−8.3, −5.0], $P_{adj} = 3.5 \times 10^{-15}$).

We then evaluated whether the main non-heritable determinants of DNA methylation variation in our cohort, i.e., age, sex, CMV serostatus, smoking status and chronic low-grade inflammation (CRP levels; Fig. 1b, Supplementary Fig. 3 and Supplementary Notes), can affect 5mC levels in a genotype-dependent manner. We thus tested for genotype × age, genotype × sex, genotype × smoking jointly (Methods). Genotype × CRP levels interactions were tested in separate models that also include the other interaction terms. We found statistical evidence for genotype-dependent effects of age and sex at 68 and 20 CpG sites, respectively ($P_{adj} < 0.05$, MAF > 0.10; Fig. 4d and Supplementary Data 3), the interacting meQTL variant being local in all cases. We detected a strong genotype × age interaction for two CpG sites located in the *BACE2* gene, the 5mC levels of which decrease with age only in donors carrying the nearby rs2837990 G > A allele (β scale 95% CI: [0.11, 0.13], $P_{adj} = 7.28 \times 10^{-10}$; Fig. 4f). *BACE2* encodes beta-secretase 2, one of two proteases involved in the generation of amyloid beta peptide, a critical component in the etiology of Alzheimer's disease[77]. Another strong genotype × age interaction effect was found for a CpG site upstream of *FCER1A*, encoding the high-affinity IgE receptor. *FCER1A* 5mC levels decrease with age in rs2251746 T > C carriers only (95% CI: [0.05,0.07], $P_{adj} = 8.6 \times 10^{-9}$), a variant known to control serum IgE levels[78]. Collectively, our analyses identify few, albeit strong, environment- and cell-type-dependent meQTLs, supporting the relatively limited impact of gene × cell type and gene × environment interactions on the blood DNA methylome.

## Cellular composition and genetics drive DNA methylation variation in human blood

Having established how cellular composition, intrinsic factors, genetic variation, and a broad selection of non-heritable factors shape the blood DNA methylome, we next sought to compare the relative impact of these factors on DNA methylation. We classified the factors into four groups: (i) the cellular composition group, which consists of the 16 measured cell proportions; (ii) the intrinsic group, which consists of age and sex; (iii) the genetic group, which consists of the most associated local-meQTL variant around each CpG site; and (iv) the exposure group, which consists of smoking status, CMV serostatus and CRP levels. Since these groups vary in their degrees of freedom, we measured the relative predictive strength for each CpG site by the out-of-sample prediction accuracy, estimated by cross-validation (Methods). To ensure unbiased estimates, we mapped local meQTLs anew within each training set.

The full model that includes all groups explains <5% of out-of-sample variance for 52.3% of CpG sites (Fig. 5a), which are typically characterized by low total 5mC variance (Supplementary Fig. 10). This suggests that these sites are constrained in the healthy population and that small fluctuations in 5mC levels determine their variation, possibly due to measurement errors or biological noise. Nevertheless, the model explains >25% of DNA methylation variance for 20.8% of CpG sites ($n = 134,305$). The strongest predictor for these CpG sites is cellular composition, genetics, intrinsic factors and exposures in 74.7%, 21.5%, 3.8% and 0.01% of cases, respectively. Cellular composition explains >25% of out-of-sample variance for 1.0% of CpG sites ($n = 90,033$; Fig. 5a, c and Supplementary Data 7), with the highest variance explained by cellular composition for one CpG site being 71.8%. For the 2,580 CpG sites where the model explains >75% of variance, local DNA sequence variation is the strongest predictor in 99.2% of cases (Fig. 5c and Supplementary Data 7). Local genetic variation explains >25% of DNA methylation variance at 23,677 CpG sites, and almost as many when adjusting for cellular composition ($n = 22,865$) (Fig. 5a, b), indicating that genetic effects on 5mC levels are mainly cell-composition-independent. Intrinsic factors explain >25% of out-of-sample variance at 3669 CpG sites, and >75% at 16 sites (Fig. 5c). When conditioning on cell composition, these numbers dropped to 334 and 6 CpG sites, respectively, suggesting that the predictive ability of age and sex is partly mediated by immune cell composition (Fig. 5b). Interestingly, environmental exposures are the weakest predictor of 5mC levels, explaining >25% of the variance at only 29 CpG sites and with a maximum variance explained for a CpG site of 50.1%.

Finally, we estimated the proportion of variance explained by genotype × age, genotype × sex and genotype × exposure interactions, by considering the difference of the out-of-sample variance explained by models including interaction terms and models with only main effects (Methods). We found a significant increase in predictive ability when including interaction terms for 431 CpG sites (ANOVA $P_{adj} < 0.05$). However, the effects were typically modest: only 13 CpG sites showed an increase in the proportion of variance explained larger than 5% (Fig. 5b). Collectively, these results show that cellular composition and local genetic variation are the main drivers of DNA methylation variation in the blood of adults, reinforcing the critical need to study epigenetic risk factors and biomarkers of disease in the context of these factors.

## Discussion

Here, we present a rich data resource that delineates the contribution of blood cellular composition, age, sex, genetics, environmental exposures, and their interactions to variation in the DNA methylome. All the results can be explored via a web-based browser (http://mimeth.pasteur.fr/), to facilitate the exploration of the estimated effects of these factors on DNA methylation variation. We found that CMV infection elicits substantial changes in the blood DNA

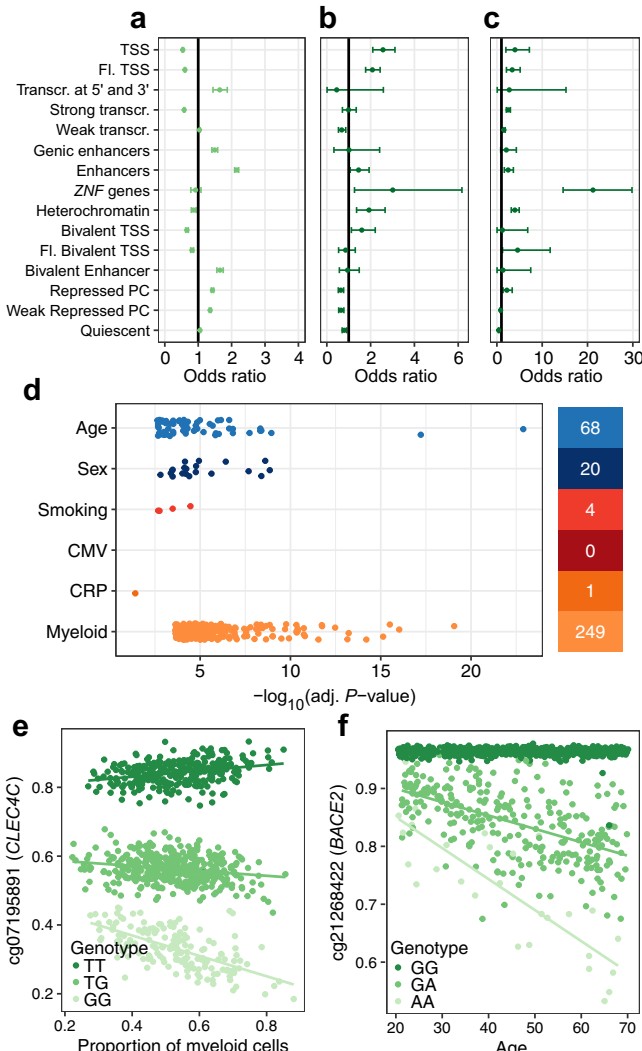

**Fig. 4 | Effects of genetics and gene × environment interactions on the blood DNA methylome. a** Enrichment in CpG sites associated with local meQTL variants, across 15 chromatin states. **b** Enrichment in CpG sites associated with remote meQTL variants, across 15 chromatin states. **c** Enrichment in remote meQTL variants, across 15 chromatin states. **d** P-value distributions for significant effects of genotype × age, genotype × sex, genotype × smoking, genotype × CMV serostatus, genotype × CRP levels and genotype × cell-type interactions. The number of significant associations is indicated on the right. Associations were tested by two-sided Wald tests with heteroscedasticity-consistent standard errors estimated by the sandwich R package[117]. Multiple testing was done by the Bonferroni correction separately for each term. **e** Myeloid lineage-dependent effect of the rs11055602 variant on 5mC levels at the *CLEC4C* locus. **f** Age-dependent effect of the rs2837990 variant on 5mC levels at the *BACE2* locus. **a–c** The point and error bars indicate the odds-ratio and 95% CI. CIs were estimated by the Fisher's exact method. Chromatin states were defined in PBMCs[15]. TSS, Fl. and PC denote transcription start site, flanking and Polycomb, respectively. **e, f** 5mC levels are given in the β value scale. Solid lines indicate linear regression lines. Statistics were computed based on a sample size of n = 884 and for 644,517 CpG sites.

methylome, in contrast with other herpesviruses such as EBV, HSV-1, HSV-2, and VZV. Latent CMV infection is known to profoundly alter the number, activation status and transcriptional profiles of immune cell populations, yet its epigenetic consequences have attracted little attention. We observed that most CMV effects on DNA methylation are mediated by the profound changes in blood cell composition[25], including the CMV-driven inflation of memory CD4+ and CD8+ T cells[33]. However, we also detected cell-composition-independent effects of CMV infection, suggesting that the herpesvirus can directly regulate

the host epigenome. Notably, differentially methylated CpG sites in CMV+ donors are strongly enriched in binding sites for BRD4, a key host regulator of CMV latency[43], suggesting that the recruitment of BRD4 by CMV during latent infection affects BRD4-regulated host genes. Furthermore, CMV+ donors are characterized by a strong increase in 5mC levels at *LTBP3*, the product of which is involved in TGF-β secretion. TGF-β is a well-known immunosuppressive cytokine induced by CMV infection[42], which represents a possible strategy of the virus to escape host immunity. These results suggest that the capacity of CMV to manipulate the host epigenetic machinery results in epigenetic changes of latently infected cells.

Our study provides further support to the notion that three different biological mechanisms underlie age-related changes in DNA methylation. The first elicits an increased dispersion of 5mC levels with age that is related to epigenetic drift[51,59–61]. We found that dispersion of DNA methylation with age is not due to cellular heterogeneity, supporting instead the progressive decline in fidelity of the DNA methylation maintenance machinery across cell populations. The second mechanism results in cell-composition-independent, global DNA demethylation and CGI-associated hypermethylation. Age-associated DNA demethylation could be related to the downregulation of DNMT3A/B de novo methyltransferases, whereas CGI-associated hypermethylation may result from the downregulation of the Polycomb repressive complexes 1 and 2 and/or TET proteins, coupled with a loss of H3K27me3 marks[79–81]. Alternatively, these changes may be related to the mitotic clock, which assumes a progressive accumulation of DNA methylation changes with mitotic divisions, including loss of methylation at partially methylated domains (PMD) and gain of methylation at PRC2-marked CpG-rich regions[82–84]. Both scenarios are supported by the enrichment of Polycomb-repressed regions in age-associated CpG sites, and of binding sites of PRC-related TFs in CpG sites methylated with age. The third mechanism elicits cell-composition-mediated demethylation at all compartments of the epigenome, particularly at enhancers of myeloid activation genes. This process likely reflects an increased degree of differentiation in the lymphoid compartment with age. Single-cell methylomes of differentiating and dividing white blood cells will help determine the role of mitotic and post-mitotic 5mC changes during epigenetic aging.

Another interesting finding of our study is that environmental exposures explain a small fraction of the variance of DNA methylation in healthy adults, at odds with the common view that the epigenome is strongly affected by the environment[85]. Twin studies have estimated the heritability of DNA methylation to range from ~20–40% (ref. 86–88), suggesting that environmental effects, along with gene × environment interactions, account for the remaining 60–80% (ref. 89). However, other factors, including cellular composition and measurement error, may account for most of the unexplained variance. Consistently, we estimated that cellular composition explains >25% of the variance for ~13% of the DNA methylome, and it has been estimated that measurement error may explain >50% (ref. 90). Nevertheless, a limitation of our study is that perinatal and early life exposures, which are thought to contribute extensively to epigenetic variation in adulthood[85], have not been extensively assessed in the Milieu Intérieur cohort. In addition, it has been hypothesized that gene × environment interactions are central to understand the role of epigenetics in development[91], but statistical evidence for interaction effects requires larger cohorts[92], suggesting that our results might represent a small, perceptible fraction of a large number of weak effects[93,94]. Large, longitudinal cohorts addressing the developmental origins of disease are needed to shed new light on the role of DNA methylation in the interplay between genes and the environment.

Collectively, our findings have broad consequences for the study and interpretation of epigenetic factors involved in disease risk. First, our analyses show that first-generation cell mixture deconvolution

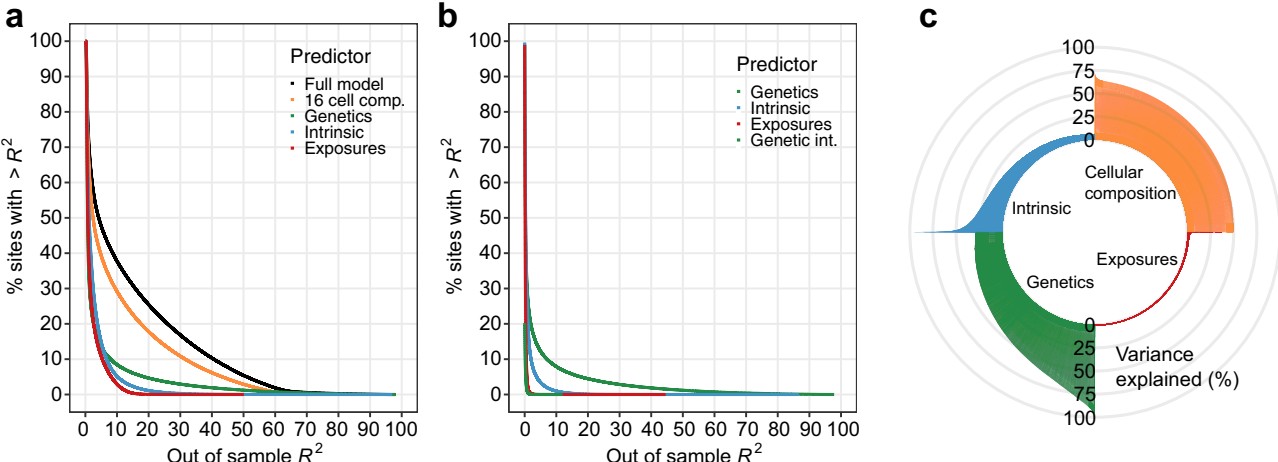

**Fig. 5 | Best predictors of the blood DNA methylome of adults. a** Complementary cumulative distribution function of the out-of-sample variance explained by the full model, blood cell composition, genetic factors, intrinsic factors (i.e., age and sex) and environmental exposures (i.e., smoking, CMV infection and CRP levels), for 644,517 CpG sites. **b** Complementary cumulative distribution function of the out-of-sample variance explained by genetic factors, intrinsic factors, environmental exposures and gene × environment (G × E) interactions, when conditioning on blood cell composition, for 644,517 CpG sites. **c** Proportion of the explained out-of-sample variance of 5mC levels for the 20,000 CpG sites with the variance most explained by blood cell composition, genetic factors, intrinsic factors and environmental exposures, respectively.

methods[18,32] do not fully distinguish direct from cell-composition-mediated effects of CMV infection and age on DNA methylation, probably because these two factors alter the proportions of blood cell subsets that are not estimated by these methods. This reinforces the view that EWAS must be interpreted with great caution, particularly when the studied diseases or conditions are known to affect unmeasured immune cell fractions. Encouragingly, our findings suggest that, when blood cell composition is not measured directly, high-resolution cell mixture deconvolution methods[29,95] provide a more complete correction for cellular heterogeneity and are therefore expected to improve the interpretation of future epigenomic studies. Second, because age, sex, CMV infection, smoking, and chronic low-grade inflammation can influence disease risk[45,96–99], our results emphasize the critical need to consider such factors in EWAS, as these factors can confound associations. Lastly, our findings reveal the epigenetic impact of aging and persistent viral infection through fine-grained changes in blood cell proportions, highlighting the need to assess the respective role of altered cellular composition and DNA methylation in the etiology of disease[17]. Large-scale studies using single-cell approaches will help overcome these challenges, and are anticipated to further decode the epigenetic mechanisms underlying healthy aging and the environmental causes of human disease.

## Methods

### The Milieu Intérieur cohort
The Milieu Intérieur cohort was established with the goal to identify genetic variation and environmental exposures that affect phenotypes related to the immune system in the adult, healthy population. The 1000 healthy donors of the Milieu Intérieur cohort were recruited by BioTrial (Rennes, France), and included 500 women and 500 men. All subjects provided written informed consent, including for genetic studies, prior to enrollment in the study. Donors included 100 women and 100 men from each decade of life, between 20 and 69 years of age. Donors were selected based on various inclusion and exclusion criteria that are detailed elsewhere[24]. Briefly, donors were required to have no history or evidence of severe/chronic/recurrent pathological conditions, neurological or psychiatric disorders, alcohol abuse, recent use of illicit drugs, recent vaccine administration, and recent use of immune modulatory agents. To avoid the influence of hormonal fluctuations in women, pregnant and peri-menopausal women were not included. To avoid genetic stratification in the study population, the recruitment of donors was restricted to individuals whose parents and grandparents were born in Metropolitan France.

### Ethical approvals
The study is sponsored by the Institut Pasteur (Pasteur ID-RCB Number: 2012-A00238-35) and was conducted as a single center study without any investigational product. The Milieu Intérieur clinical study was approved by the Comité de Protection des Personnes−Ouest 6 (Committee for the protection of persons) on June 13, 2012 and by the French Agence Nationale de Sécurité du Médicament (ANSM) on June 22, 2012. The samples and data used in this study were formally established as the Milieu Intérieur biocollection (study# NCT03905993), with approvals by the Comité de Protection des Personnes−Sud Méditerranée and the Commission nationale de l'informatique et des libertés (CNIL) on 11 April 2018.

### DNA sampling and extraction
Whole blood was drawn from the 1000 Milieu Intérieur healthy, fasting donors every working day from 8AM to 11AM, from September 2012 to August 2013, in Rennes, France. Different anticoagulants were used, depending on the downstream analyses. For DNA methylation profiling, blood samples were collected on EDTA, whereas samples for flow cytometry and genome-wide DNA genotyping were collected on Li-heparin. Tracking procedures were established in order to ensure delivery to Institut Pasteur (Paris) within 6 h of blood draw, at a temperature between 18 °C and 25 °C. Upon receipt, samples were kept at room temperature until DNA extraction. DNA for DNA methylation profiling was extracted using the Nucleon BACC3 genomic DNA extraction kit (catalog #: RPN8512; Cytiva, Massachusetts, USA). High-quality genomic DNA was obtained for 978 out of the 1000 donors.

### DNA methylation profiling and data quality controls
Extracted genomic DNA was treated with the EZ DNA Methylation Kit (catalog #: D5001; Zymo Research, California, USA). Bisulfite-converted DNA was applied to the Infinium MethylationEPIC Bead-Chip (catalog #: WG-317–1003; Illumina, California, USA), using the manufacturer's standard conditions. The MethylationEPIC BeadChip measures 5mC levels at 866,836 CpG sites in the human genome. Raw IDAT files were processed with the minfi R package[100]. All samples

showed average detection *P*-values < 0.005. No sample showed a mean of methylated intensity signals lower than 3 × standard deviations (SD) from the cohort average. Therefore, no samples were excluded based on detection *P*-values or methylated intensity signals. The sex predicted from 5mC signals on sex chromosomes matched the declared sex for all samples (Supplementary Fig. 1a). Using the 59 control SNPs included in the MethylationEPIC array, a single sample showed high genotype discordance with the genome-wide SNP array data (see 'Genome-wide DNA genotyping' section) and was thus excluded (Supplementary Fig. 1b). Unmethylated and methylated intensity signals were converted to *M*-values. A total of 2930 probes with >1% missingness (i.e., detection *P*-value > 0.05 for more than 1% of donors) were excluded and remaining missing data (missingness = 0.0038%) were imputed by mean substitution. Using the irlba R package, Principal Component Analysis (PCA) of M values identified nine outlier samples, including eight that were processed on the same array (Supplementary Fig. 1c), which were also excluded. The "noob" background subtraction method[101] was applied on M values for the remaining 968 samples, which showed highly consistent epigenome-wide DNA methylation profiles (Supplementary Fig. 1d, e).

To identify batch effects on the DNA methylation data, we searched for the factors that were the most associated with the top 20 PCs of the PCA of noob-corrected M values. We used a linear mixed model that included age, sex and cytomegalovirus (CMV) serostatus as fixed effects, and slide position and sample plate as random effects. The models were fitted with the lme4 R package[102]. Strong associations were observed between the first four PCs and slide position and sample plate (Supplementary Fig. 1f, g). M values were thus corrected for these two batch effects using the ComBat function, from the sva R package[103]. After ComBat correction, the ten first PCs of a PCA of M values were associated with factors known to affect DNA methylation, including blood cell composition, age and sex (Supplementary Fig. 1h–j), indicating no other, strong batch effect on the data (see section 'Associations with principal components of DNA methylation').

M-values were converted to β values, considering that $\beta = 2^M / (2^M + 1)$. Because outlier 5mC values due to measurement error could inflate the type I error rate of regression models, we excluded, for each CpG site, M or β values that were greater than 5 × SD from the population average, corresponding to <0.1% of all measures. We also excluded (i) 83,380 non-specific probes that share >90% sequence identity with several genomic regions (see details in[104]), (ii) 118,575 probes that overlap a SNP that is within the 50 pb surrounding the CpG site and has a MAF > 1% in the Milieu Intérieur cohort or in European populations from the 1000 Genomes project[105], (iii) 558 probes that were absent from the Illumina annotations version 1.0 B4 and (iv) 16,876 probes located on sex chromosomes. As a result, the final, quality-controlled data was composed of 968 donors profiled at 644,517 CpG sites.

## Flow cytometry

Immune cell proportions were measured using ten eight-color flow-cytometry panels[25]. The acquisition of cells was performed using two MACSQuant analyzers, which were calibrated using MacsQuant calibration beads (Miltenyi, Germany). Flow cytometry data were generated using MACSQuantify software version 2.4.1229.1. The mqd files were converted to FCS compatible format and analyzed by FlowJo software version 9.5.3. A total of 110 cell proportions were exported from FlowJo. Protocols, panels, staining antibodies, and quality control filters used for flow cytometry analyses are detailed elsewhere[25]. Abnormal lysis or staining were systematically flagged by trained experimenters. We removed outliers by using a scheme detailed previously[25]. We used a distance-based approach that, for each cell type, removes observations in the right tail if the distance to the closest observation in the direction of the mean is larger than 20% of the range of the observations. Similarly, observations in the left tail were

removed if the distance to the closest observation in the direction of the mean is more than 15% than the range the observations. We removed 22 observations in total, including a maximum of 8 observations for a single cell type (i.e., for the proportion of neutrophils). Problems in flow cytometry processing, such as abnormal lysis or staining, were systematically flagged by trained experimenters, which resulted in 8.7% missing data. Because imputing missing data for donors who show large missingness could be inaccurate, we excluded 74 donors with no data for the T cell panel. Finally, the remaining missing data were imputed using the random forest-based missForest R package[106].

## Genome-wide DNA genotyping

The 1000 Milieu Intérieur donors were genotyped on both the HumanOmniExpress-24 and the HumanExome-12 BeadChips (Illumina, California, USA), which include 719,665 SNPs and 245,766 exonic SNPs, respectively. Average concordance rate between the two genotyping arrays was 99.9925%. The combined data set included 732,341 high-quality polymorphic SNPs. After genotype imputation and quality-control filters[25], a total of 11,395,554 SNPs was further filtered for minor allele frequencies >5%, yielding a data set composed of 1000 donors and 5,699,237 SNPs for meQTL mapping. Ten pairs of first to third-degree related donors were detected with KING 1.9 (ref. [107]). Out of the 894 donors whose blood methylome and blood cell composition were accurately profiled, 884 unrelated donors were kept for subsequent analyses.

## Immune cell proportions

One of the key questions in this study is whether differences in 5mC levels observed with respect to different factors are due to epigenetic changes occurring within cell types or if they in fact reflect changes in blood cell composition. To answer this question, we considered the proportions of 16 major subsets of blood: naïve, central memory (CM), effector memory (EM) and terminally differentiated effector memory (EMRA) subsets of CD4$^+$ and CD8$^+$ T cells, CD4$^-$CD8$^-$ T cells, B cells, dendritic cells, natural killer (NK) cells, monocytes, neutrophils, basophils and eosinophils[25]. As these cellular proportions were measured by flow cytometry using a hierarchical gating strategy[25], they are expected to sum to one. Yet, because of measurement errors, cell fractions do not exactly sum to one in all donors. For a measure of the proportion of a given cell subset in a given donor, we therefore used the absolute count of the cell type divided by the sum of absolute counts of all the 16 measured cell subsets. We used the same approach when considering a reduced set of six major cell types, comprising neutrophils, monocytes, NK cells, B cells, and CD4$^+$ and CD8$^+$ T cells, for comparison purposes.

## Compositional analysis of cellular composition

We sought to study the association between 5mC levels and blood cell composition, experimentally measured by flow cytometry. However, the 16 measured cellular proportions are constrained to be positive and to sum to one. Consequently, a change in one cellular proportion must necessarily change one or more of the other cellular proportions, complicating the interpretation of parameters estimated from linear regression models with measured immune cell proportions as predictors[28,108,109]. Here, we investigated instead the effect of balances, which are transformations of cell-type proportions that can be seen as a generalization of the logit-transform. These balances model the effect of a relative change between two groups of cell types. They are defined in a hierarchical manner of increasing granularity, by a sequential binary partition (SBP) of the 16 measured cell types, generating 15 balances in total (Supplementary Data 2). As an example, we describe the first two balances. The other balances are defined in an analogous manner according to the SBP and the general procedure detailed elsewhere[108]. The first balance captures the relative effect on

5mC levels of the myeloid cell types compared to the lymphoid cell types. Of the 16 measured cell types, five are myeloid and eleven are lymphoid. Let $c_i^{M_1}, \ldots, c_i^{M_5}$ be the measured myeloid proportions and $c_i^{L_1}, \ldots c_i^{L_{11}}$ be lymphoid proportions for the $i$:th individual. The first balance predictor for that individual is defined by

$$b_i^1 = \sqrt{\frac{5 \times 11}{5 + 11}} \log \left\{ \frac{\prod_{m=1}^{5} c_i^{M_m}}{\prod_{l=1}^{11} c_i^{L_l}} \right\}, \tag{1}$$

The second balance is defined within the lymphoid group and captures the relative effect on 5mC levels of T cells with respect to NK cells and B cells. Let $c_i^{T_1}, \ldots, c_i^{T_9}$ be the measured proportions of the nine types of T cells and let $c_i^B$ and $c_i^{NK}$ be proportions of B cells and NK cells. The balance contrasting T cells with NK cells and B cells is given by

$$b_i^2 = \sqrt{\frac{9 \times 2}{9 + 2}} \log \left\{ \frac{\prod_{m=1}^{9} c_i^{T_m}}{c_i^B c_i^{NK}} \right\}. \tag{2}$$

All balances were computed from the SBP using the robCompositions R package[110]. To evaluate the validity of our approach, we compared the estimated effects on 5mC levels of balances contrasting two groups of cell-types with the measured differences in 5mC levels between the same two groups, obtained from MethylationEPIC data in sorted cell-types[29] and found strong correlations ($R > 0.6$; Supplementary Fig. 2 and Supplementary Data 2). We further evaluated the accuracy of our approach by performing a simulation study. First, we simulated 5mC levels based on observed cell composition data and evaluated how the balances capture 5mC differences in the relevant cell types. Second, we simulated cell composition data from a Dirichlet distribution and again evaluated that regression models including the balances as predictors give the expected results (Supplementary Notes).

The 15 balances were used to investigate the effects of immune cell composition on 5mC levels at individual CpG sites (see section 'Epigenome-wide association study of cell composition') and on principal components of epigenome-wide DNA methylation levels (see section 'Associations with principal components of DNA methylation').

**Epigenome-wide association study of cell composition**
To investigate how immune cell composition affects the blood DNA methylome, we investigated effects of cell-type balances on 5mC levels at each CpG site. For the $p$:th CpG site and the $i$:th individual, introduce observed 5mC levels $y_i^p$ measured on the M value scale. Let $\mathbf{b}_i$ be a vector of 15 cell-type balances with corresponding parameter vector $\boldsymbol{\beta}_b^p$. Let the vector $\mathbf{SNP}_i^p$ contain the significant local SNP with the smallest $P$-value and all independently associated remote SNPs (see section 'Local meQTL mapping analyses' and section 'Remote meQTL mapping analyses') with corresponding parameter vector $\boldsymbol{\beta}_{SNP}^p$. We performed an epigenome-wide association analysis of cellular composition by fitting the models,

$$y_i^p = \mu^p + \mathbf{b}_i^t \boldsymbol{\beta}_b^p + \left(\mathbf{SNP}_i^p\right)^t \boldsymbol{\beta}_{SNP}^p + \varepsilon_i^p, \tag{3}$$

where $\varepsilon_i^p \sim \left(0, \sigma_p^2\right)$. Models were fitted by ordinary least squares. For each balance in $\mathbf{b}_i$ (see Eqs. (1) and (2) for examples), the parameters in $\boldsymbol{\beta}_b^p$ are interpreted as the change in 5mC levels for an increase in the first cell-type group and the corresponding decrease in the second cell-type group.

**Associations with principal components of DNA methylation**
To evaluate how principal components (PCs) of DNA methylation levels are related to cell composition, we first computed PCs of 5mC levels at 644,517 CpG sites, with the irlba R package. Let $y_i^k$ be the observed value of the $k$:th PC of the DNA methylation data and $\mathbf{b}_i$ a

vector of 15 cell-type balances measured for individual $i$ with the corresponding parameter vector $\boldsymbol{\beta}_b^k$. Given that we observed variability in 5mC levels across dates of blood draw, we included them as random effects. Let $j$ be the day of blood draw for the $i$:th individual. The model we used to estimate the effects of cellular composition on PCs of DNA methylation was,

$$y_i^k = \mu^k + \mathbf{b}_i^t \boldsymbol{\beta}_b^k + \text{DateOfSampling}_{j(i)}^k + \varepsilon_i^k, \tag{4}$$

with $\text{DateOfSampling}_{j(i)}^k \sim \mathcal{N}\left(0, \tau_k^2\right)$ and $\varepsilon_i^k \sim \left(0, \sigma_k^2\right)$. The models were fitted with the lme4 R package[102].

To evaluate how PCs of DNA methylation levels are related to the candidate non-heritable factors, i.e., age, sex, smoking status, CMV serostatus, introduce the variables $\text{Age}_i$, $\text{Woman}_i$, $\text{Exsmoker}_i$, $\text{Smoker}_i$ and $\text{CMV}_i$ with corresponding parameters $\beta_{Age}^k$, $\beta_{Woman}^k$, $\beta_{Exsmoker}^k$, $\beta_{Smoker}^k$ and $\beta_{CMV}^k$. Let $\text{PC1}_i$ and $\text{PC2}_i$ be the two first PCs of the genotype matrix. Let $\mathbf{c}_i$ be a vector of 15 measured cell proportions, excluding neutrophils because of the sum-to-one constraint, and $\boldsymbol{\beta}_c^k$ the corresponding parameter vector. The model we used to estimate the effects of non-genetic factors on PCs of DNA methylation was,

$$
\begin{aligned}
y_i^k = \mu^k &+ \mathbf{c}_i^t \boldsymbol{\beta}_c^k + \text{Age}_i \beta_{Age}^k + \text{Woman}_i \beta_{Woman}^k + \text{Exsmoker}_i \beta_{Exsmoker}^k \\
&+ \text{Smoker}_i \beta_{Smoker}^k + \text{CMV}_i \beta_{CMV}^k + \text{PC1}_i \beta_{PC1}^k + \text{PC2}_i \beta_{PC2}^k \\
&+ \text{DateOfSampling}_{j(i)}^k + \varepsilon_i^k.
\end{aligned}
\tag{5}
$$

The models were fitted with the lme4 R package[102]. Inference was performed using the Kenward-Roger $F$-test approximation for linear mixed models, implemented in the pbkrtest R package[111].

**Epigenome-wide association studies of non-genetic factors**
We assessed the effects of 141 non-genetic variables (Supplementary Data 1) on the blood DNA methylome of adults. The measured 5mC levels at a CpG site are the average of the DNA methylation state at this CpG site of all cells in the blood sample. Many of the 141 candidate variables might influence cell composition, which will cause a corresponding change in 5mC levels. We denote this effect the "(cell-composition-)mediated effect". In addition, the variable might alter 5mC levels within individual cells, or within cell-types. We denote this effect the direct effect (see Supplementary Fig. 11 for a schematic directed acyclic graph of the system). Several factors are known to have a large effect on blood cell composition in healthy donors, the most important being age, sex, CMV serostatus and smoking[25]. As an added complexity, these factors are also associated with most of the other variables in the study. Based on this framework, we investigated four questions, each one targeted by a separate statistical model.

**The total effect.** The total effect includes both changes in 5mC levels induced by changes in cellular composition (i.e., cell-composition-mediated effects) and those induced within cell types (i.e., direct effects). For each variable of interest $x$ and each CpG site, the total effect was estimated in a regression model including, as response variable, the 5mC levels of the CpG site on the M value scale and, as predictors, $x_i$, a nonlinear age term of 3 DoF natural splines, sex, CMV serostatus, smoking status, the significant local SNP with the smallest $P$-value, independently associated remote SNPs and the first two PCs of the genotype matrix. Again, since we observed variability in 5mC levels across dates of blood draw, we included them as a random effect term. For the $p$:th CpG site, let $y_i^p$ be the 5mC levels of the $i$:th individual on the M value scale, $f_{Age}^p(\text{Age}_i)$ a nonlinear age term of 3 DoF natural splines and $\mathbf{SNP}_i^p$ a vector of the minor allele counts for the significant local SNP with the smallest $P$-value and independently associated remote SNPs, with corresponding parameter vector $\boldsymbol{\beta}_{SNP}^p$. The total effect of the variable $x_i$ was estimated by the corresponding parameter

$\beta_x^p$ in the models,

$$
\begin{aligned}
y_i^p = {} & \mu^p + x_i\beta_x^p + f_{\text{Age}}^p(\text{Age}_i) + \text{Woman}_i\beta_{\text{Woman}}^p + \text{Exsmoker}_i\beta_{\text{Exsmoker}}^p \\
& + \text{Smoker}_i\beta_{\text{Smoker}}^p + \text{CMV}_i\beta_{\text{CMV}}^p + \text{PC1}_i\beta_{\text{PC1}}^p + \text{PC2}_i\beta_{\text{PC2}}^p \quad (6) \\
& + \left(\mathbf{SNP}_i^p\right)^t \boldsymbol{\beta}_{\mathbf{SNP}}^p + \text{DateOfSampling}_{j(i)}^p + \varepsilon_i^p,
\end{aligned}
$$

where $\text{DateOfSampling}_{j(i)}^p \sim \mathcal{N}\left(0, \tau_p^2\right)$ and $\varepsilon_i^p \sim \left(0, \sigma_p^2\right)$. The effect of aging was tested in models with $x$ removed and the non-linear age term replaced by a linear one. The effects of sex, smoking status and CMV serostatus were tested in models where we removed $x$. For variables relating to women only (e.g., age of menarche), we excluded men from the analysis and removed $\text{Woman}_i\beta_{\text{Woman}}^p$. The models were fitted with the lme4 R package[102]. Hypothesis tests were performed using the Kenward-Roger approximation of the $F$-test for linear mixed models, implemented in the pbkrtest R package[111].

**The direct effect.** Let the vector $\mathbf{c}_i$ be measured proportions of the 15 immune cell types, excluding neutrophils, for the $i$:th individual and $\boldsymbol{\beta}_c^p$ the corresponding parameter vector. Using the same notation as for the total effect, the direct effect of the variable $x_i$ was estimated by $\beta_x^p$ in the models,

$$
\begin{aligned}
y_i^p = {} & \mu^p + x_i\beta_x^p + \mathbf{c}_i^t\boldsymbol{\beta}_c^p + f_{\text{Age}}^p(\text{Age}_i) + \text{Woman}_i\beta_{\text{Woman}}^p \\
& + \text{Exsmoker}_i\beta_{\text{Exsmoker}}^p + \text{Smoker}_i\beta_{\text{Smoker}}^p + \text{CMV}_i\beta_{\text{CMV}}^p + \text{PC1}_i\beta_{\text{PC1}}^p \\
& + \text{PC2}_i\beta_{\text{PC2}}^p + \left(\mathbf{SNP}_i^p\right)^t \boldsymbol{\beta}_{\mathbf{SNP}}^p + \text{DateOfSampling}_{j(i)}^p + \varepsilon_i^p,
\end{aligned}
$$
$$(7)$$

We also tested the interaction effect of sex, CMV serostatus and smoking status with age by including one interaction term at a time in the model specified in Eq. (7). The models were fitted with the lme4 R package[102]. Hypothesis tests were performed by the Kenward-Roger approximation of the $F$-test for linear mixed models, implemented in the pbkrtest R package[111].

**The mediated effect.** The cell-composition-mediated effect was estimated as the effect on 5mC levels mediated by changes in proportions of the 16 cell subsets due to the given factor. We estimated the mediated effect of aging, sex, variables related to smoking, CMV serostatus and heart rate. The mediated effect was estimated using a two-stage procedure. First, we fitted models with measured proportions of immune cells as response variables. Let $\mathbf{c}_i$ be a vector of measured proportions of the 15 blood subsets, excluding neutrophils. Let $c_i^n$ denote the $n$:th entry of the vector $\mathbf{c}_i$, i.e., the measured proportion of the $n$:th cell-type for the $i$:th individual. Introduce the vector $\mathbf{k}_i$ of covariate values for the $i$:th individual, including age (3 DoF spline with an entry for each term), sex, smoking, CMV serostatus and ancestry (2 PCs), but excluding the variable of interest $x_i$ (mediated effect of aging was estimated with a linear term). For the model of the $n$:th cell-type, let $\boldsymbol{\beta}_k^n$ be the parameter vector for the covariate vector $\mathbf{k}_i$ and $\beta_x^n$ the parameter for the variable of interest $x_i$. In the first stage, we fitted the models,

$$
E\{c_i^n | x_i, \mathbf{k}_i\} = \beta_0 + x_i\beta_x^n + \mathbf{k}_i^t\boldsymbol{\beta}_k^n, \quad n = 1, \dots, 15. \quad (8)
$$

Next, let $y_i^p$ be 5mC levels in the M value scale for the $p$:th CpG site, $\theta_x^p$ the parameter for the variable of interest, and $\boldsymbol{\theta}_c^p$ and $\boldsymbol{\theta}_k^p$ parameter vectors for the effects of cell proportions and covariates. In the second stage, we fitted the models,

$$
E\{y_i^p | x_i, \mathbf{c}_i, \mathbf{k}_i\} = \theta_0^p + x_i\theta_x^p + \mathbf{c}_i^t\boldsymbol{\theta}_c^p + \mathbf{k}_i^t\boldsymbol{\theta}_k^p. \quad (9)
$$

The mediated effect of $x_i$ on DNA methylation was estimated by $\boldsymbol{\beta}_x^t\boldsymbol{\theta}_c^p$ (ref. 34). Inference was performed by the parametric bootstrap.

**The direct effects adjusted by deconvolution methods.** To compute the IDOL and Houseman-adjusted effects, we estimated proportions of CD4$^+$ and CD8$^+$ T cells, B cells, NK cells, monocytes, and neutrophils, using the estimateCellCounts2 function in the Flow-Sorted.Blood.EPIC package with either Houseman et al.'s CpG sites, or IDOL optimized CpG sites[112]. For age, sex, smoking status, CMV serostatus, heart rate, ear temperature and hour of blood draw, we estimated the IDOL- and Houseman-adjusted effect by adjusting for estimated 5 proportions in the model specified by Eq. (7), instead of the 15 measured proportions, excluding neutrophils because of the sum-to-one constraint. To compute the EPIC IDOL-Ext-adjusted effects, we estimated proportions of 12 major cell types in blood, including CD4$^+$ and CD8$^+$ T cells, naïve and differentiated subtypes of CD4$^+$ and CD8$^+$ T cells, neutrophils, monocytes, basophils, eosinophils, NK cells, regulatory T cells, naïve and memory B cells, using the IDOL-Ext reference matrix in the estimateCellCounts2 function from the FlowSorted.BloodExtended.EPIC R package[29]. We estimated the IDOL-Ext-adjusted effect by including 11 estimated proportions in Eq. (7) instead of the 15 measured proportions, excluding neutrophils because of the sum-to-one constraint. Finally, for comparison purposes, we also computed the association between non-genetic factors and 5mC levels by adjusting, in Eq. (7), for the proportions of the 5 major cell types measured by flow cytometry, instead of the 15 measured proportions, excluding again neutrophils.

## Prediction of CMV serostatus

We built a prediction model to estimate CMV serostatus from DNA methylation data using elastic net regression for binary data[113], implemented in the glmnet R package[114]. We included all CpG sites as predictors in the model, including those on the X and Y chromosomes. The model was built from 863,906 CpG sites in 969 samples. The elastic net model has two tuning parameters that determine the degree of regularization of the predictor function. We selected both tuning parameters by two-dimensional five times repeated cross-validation over the two parameters. The final model fitted on the full data set includes 547 CpG sites with non-zero parameters.

## Detection of the dispersion of DNA methylation with age

To estimate changes in dispersion of 5mC levels with age, we fitted regression models where the residual variance depends on age. Let $y_i^p$ be 5mC levels on the M value scale for the $p$:th CpG site and the $i$:th individual. Using similar notations as above, we estimated the dispersion effect of age by the parameter $\theta^p$ in the models,

$$
\begin{aligned}
y_i^p = {} & \mu^p + \mathbf{c}_i^t\boldsymbol{\beta}_c^p + \left(\mathbf{SNP}_i^p\right)^t \boldsymbol{\beta}_{\mathbf{SNP}}^p + f_{\text{Age}}^p(\text{Age}_i) + \text{Woman}_i\beta_{\text{Woman}}^p \\
& + \text{Exsmoker}_i\beta_{\text{Exsmoker}}^p + \text{Smoker}_i\beta_{\text{Smoker}}^p + \text{CMV}_i\beta_{\text{CMV}}^p \quad (10) \\
& + \text{PC1}_i\beta_{\text{PC1}}^p + \text{PC2}_i\beta_{\text{PC2}}^p + \varepsilon_i^p,
\end{aligned}
$$

where

$$
\varepsilon_i^p \sim \mathcal{N}\left(0, \sigma_{i,p}^2\right), \log\sigma_{i,p} = \tau^p + \text{Age}_i\theta^p. \quad (11)
$$

We devised a hypothesis test for $\theta$ by a likelihood ratio test comparing the model in Eq. (11), to a model with

$$
\varepsilon_i^p \sim \mathcal{N}\left(0, \sigma_p^2\right), \log\sigma_p = \tau^p. \quad (12)
$$

As a sensitivity analysis, we also fitted a model with

$$
\varepsilon_i^p \sim \mathcal{N}\left(0, \sigma_{i,p}^2\right), \log\sigma_{i,p} = \tau^p + \text{Age}_i\theta^p + \mathbf{c}_i^t\boldsymbol{\beta}_c^p. \quad (13)
$$

In this case, the hypothesis test for $\theta$ was done by comparing to a model with

$$\varepsilon_i^p \sim \mathscr{N}\left(0, \sigma_{i,p}^2\right), \log\sigma_{i,p} = \tau^p + \mathbf{c}_i^t\boldsymbol{\beta}_c^p. \qquad (14)$$

These models were fitted with the gamlss R package[115].

### Local meQTL mapping analyses

Local meQTL mapping was performed using the MatrixEQTL R package[116]. Association was tested for each CpG site and each SNP in a 100-Kb window around the CpG site, by fitting a linear regression model assuming an additive allele effect. Models included, as predictors, the 15 immune cell proportions, a nonlinear age term encoded by 3 degrees-of-freedom (DoF) natural splines, sex, smoker status, ex-smoker status and CMV serostatus. We also adjusted for the top two PCs of a PCA of the genotype data. We did not include more PCs because of the low population substructure observed in the cohort[25]. For the $i$:th individual and the $p$:th CpG site, let $y_i^p$ be the measured 5mC levels on the M value scale, $\mathrm{SNP}_i^{p,m}$ the minor allele count of the $m$:th tested SNP for the CpG site and $f_{\mathrm{Age}}^{p,m}(\mathrm{Age}_i)$ a nonlinear age term of natural splines. Moreover, let the vector $\mathbf{c}_i$ be measured proportions of the 15 immune cell-types for the $i$:th individual, excluding neutrophils, and $\boldsymbol{\beta}_c^{p,m}$ the corresponding parameter vector. The additive allele effect of the SNP was estimated by the parameter $\beta_m^{p,m}$ in the models,

$$
\begin{aligned}
y_i^p = {}& \mu^{p,m} + \mathrm{SNP}_i^{p,m}\beta_m^{p,m} + f_{\mathrm{Age}}^{p,m}(\mathrm{Age}_i) + \mathrm{Woman}_i\beta_{\mathrm{Woman}}^{p,m} \\
& + \mathrm{Exsmoker}_i\beta_{\mathrm{Exsmoker}}^{p,m} + \mathrm{Smoker}_i\beta_{\mathrm{Smoker}}^{p,m} \\
& + \mathrm{CMV}_i\beta_{\mathrm{CMV}}^{p,m} + \mathrm{PC1}_i\beta_{\mathrm{PC1}}^{p,m} + \mathrm{PC2}_i\beta_{\mathrm{PC2}}^{p,m} + \mathbf{c}_i^t\boldsymbol{\beta}_c^{p,m} + \varepsilon_i^{p,m},
\end{aligned} \qquad (15)
$$

where $\varepsilon_i^{p,m}$ is a symmetrical zero-mean distribution with constant variance.

### Remote meQTL mapping analyses

Testing all possible associations between 644,517 CpG sites and 5,699,237 SNPs would require performing 3769 billion statistical tests. To reduce the multiple testing burden, remote meQTL mapping was conducted on a selection of 50,000 CpG sites with the highest residual variance in the model described in Eq. (15), but with $m$ indexing in this case only the most associated local SNP for the $p$:th CpG site. For each of the 50,000 selected CpG sites, we then fitted one model per SNP located outside of a 1-Mb window around the CpG site. For each SNP-CpG pair, we estimated the additive allele effect of the remote SNP using the model specified in Eq. (15) but with $m$ now indexing remote SNPs for the $p$:th CpG site. Both local and remote meQTL mapping tests were corrected for multiple testing by the Bonferroni adjustment.

### Detection of independent remote meQTLs

We designed the following scheme to compute a set $\Phi$ of independently associated remote SNPs for each CpG site, where all such SNPs are associated with 5mC levels $y^p$ at the $p$:th CpG site, conditional on the most associated local SNP and other SNPs in $\Phi$. Define $X_1$ to be the set of SNPs with a remote association to $y^p$ and let $x^0$ be the most associated significant local SNP, if it exists. The set $X_1$ typically includes several SNPs that are in linkage disequilibrium (LD). The algorithm uses an iterative procedure to build sets $M_j$ of SNPs, where in the $j$:th iteration, SNPs that are not associated with 5mC levels at the CpG site conditional on SNPs included in $M_{j-1}$ are discarded, while the most associated is retained in $M_j$. In the final step, the set $\Phi$ is constructed by elements of the final set $M$ that are associated with 5mC levels at the CpG site conditional on all the other elements in $M$. Intuitively, $\Phi$ consists of the most associated SNP in each LD block. The algorithm is given in pseudocode in Algorithm (1), where the condition $\beta^p \neq 0$ is determined by an $F$-test on the level $\alpha = 10^{-6}$.

**Algorithm 1**. Forming a set of remote independently associated SNPs with a CpG site.

If the CpG site is under local genetic control then let $M_1 = x_0$, otherwise let $M_1 = \varnothing$
Repeat for $j = 1, 2, \ldots$
$P = \left\{ x \in X_j \setminus M_j : \beta_x^p \neq 0 \text{ in } y_i^p = \mu^p + x_i\beta_x^p + \sum_{z \in M_j} z_i\beta_z^p + \varepsilon_i^p, \varepsilon_i^p \sim \left(0, \sigma_p^2\right) \right\}$
If $P = \varnothing$ Exit
$X_{j+1} = P$
$M_{j+1} = M_j \cup \{x : x \text{ SNP with the smallest } P\text{-value in } P\}$
End
$\Phi = \{ x \in M_{j+1} \setminus x_0 : \beta_x^p \neq 0$
in $y_i^p = \mu^p + x_i\beta_x^p + \sum_{z \in M_{j+1}\setminus\{x\}} z_i\beta_z^p + \varepsilon_i^p, \varepsilon_i^p \sim (0, \sigma_p^2) \}$

### Cell-type-dependent effects of genetic and non-genetic factors on DNA methylation

To investigate whether the effects of a factor on DNA methylation depend on the proportion of myeloid cells in blood, we fitted models that included an interaction term between the factor of interest (i.e., age, sex, smoking status, CMV serostatus and genetic variants) and the proportion of myeloid cells, $c_i^m$, defined as the sum of the proportions of cell-types from the myeloid lineage. With the same notations as above, but with $y_i^p$ being 5mC levels on the β value scale for the $p$:th CpG site and the $i$:th individual, we estimated the cell-type-dependent effects of non-genetic factors by fitting the models,

$$
\begin{aligned}
y_i^p = {}& \mu^p + \mathrm{Age}_i\beta_{\mathrm{Age}}^p + \mathrm{CMV}_i\beta_{\mathrm{CMV}}^p + \mathrm{Woman}_i\beta_{\mathrm{Woman}}^p + \mathrm{Smoker}_i\beta_{\mathrm{Smoker}}^p \\
& + \mathrm{PC1}_i\beta_{\mathrm{PC1}}^p + \mathrm{PC2}_i\beta_{\mathrm{PC2}}^p + c_i^m\beta_{c^m}^p + c_i^m \\
& \times \left( \mathrm{Woman}_i\theta_{\mathrm{Woman}}^p + \mathrm{Age}_i\theta_{\mathrm{Age}}^p + \mathrm{Smoker}_i\theta_{\mathrm{Smoker}}^p + \mathrm{CMV}_i\theta_{\mathrm{CMV}}^p \right) + \varepsilon_i^p.
\end{aligned} \qquad (16)
$$

We also investigated whether the effect of genotypes could be dependent on the proportion of myeloid cells in the sample. For the $p$:th CpG site and the $i$:th individual, let $\mathrm{SNP}_i^{p,k}$ be the minor allele counts of the significant local SNP with the smallest $P$-value and independently associated remote SNPs. In this case, we also use 5mC levels on the β value scale. We estimated the cell-type-dependent effects of genetic factors by fitting the models,

$$
\begin{aligned}
y_i^p = {}& \mu^p + f_{\mathrm{Age}}^p(\mathrm{Age}_i) + \mathrm{CMV}_i\beta_{\mathrm{CMV}}^p + \mathrm{Woman}_i\beta_{\mathrm{Woman}}^p + \mathrm{Smoker}_i\beta_{\mathrm{Smoker}}^p \\
& + \mathrm{PC1}_i\beta_{\mathrm{PC1}}^p + \mathrm{PC2}_i\beta_{\mathrm{PC2}}^p + c_i^m\beta_{c^m}^p + \sum_k \mathrm{SNP}_i^{p,k}\beta_{\mathrm{SNP}^{p,k}} \\
& + c_i^m \left( \sum_k \mathrm{SNP}_i^{p,k}\theta_{\mathrm{SNP}^{p,k}} \right) + \varepsilon_i^p.
\end{aligned} \qquad (17)
$$

Inference in both cases was done by Wald tests with heteroscedasticity-consistent standard errors estimated by the sandwich R package[117].

### Detection of gene × environment interactions

We tested whether age, sex, CMV serostatus, smoking status or CRP levels could have a genotype-dependent effect on the DNA methylome. For the $i$:th individual and the $p$:th CpG site, let $y_i^p$ be the 5mC levels on the M value scale, $\mathrm{SNP}_i^{p,k}$, $k = 1, \ldots, K^p$, the minor allele counts of the significant local meQTL with the lowest $P$-value and the $K^p - 1$ independently associated remote meQTLs, and $\mathbf{c}_i$ the vector of 15 measured immune cell proportions with corresponding parameter vector $\boldsymbol{\beta}_c^p$. Interaction effects were estimated for each CpG site in the

model,

$$
\begin{aligned}
E\left\{y_i^p | \text{SNP}_i^{p,1}, \ldots, \text{SNP}_i^{p,K^p}, \text{Age}_i, \text{Woman}_i, \text{Smoker}_i, \text{CMV}_i\right\} \\
= \mu^p + \sum_{k=1}^{K^p} \text{SNP}_i^{p,k}\beta_{\text{SNP}^{p,k}} + \mathbf{c}_i^t\boldsymbol{\beta}_c^p + \text{PC1}_i\beta_{\text{PC1}}^p + \text{PC2}_i\beta_{\text{PC2}}^p + \text{Age}_i\beta_{\text{Age}}^p \\
+ \text{Woman}_i\beta_{\text{Woman}}^p + \text{Smoker}_i\beta_{\text{Smoker}}^p \, \text{CMV}_i\beta_{\text{CMV}}^p \\
+ \sum_{k=1}^{K^p} \text{SNP}_i^{p,k}\left(\text{Age}_i\theta_{\text{Age}}^{p,k} + \text{Woman}_i\theta_{\text{Woman}}^{p,k} + \text{Smoker}_i\theta_{\text{Smoker}}^{p,k} + \text{CMV}_i\theta_{\text{CMV}}^{p,k}\right)
\end{aligned}
$$
(18)

We investigated effects of CRP levels in a separate model that simply added a log-transformed CRP term to Eq. (18). Inference was done by Wald tests with heteroscedasticity-consistent standard errors estimated by the sandwich R package[117].

## Estimation of proportions of explained 5mC variance

According to our analyses, 5mC levels in the healthy population are mainly associated with local genetic variation, blood cell composition, age, sex, smoking, CMV infection and CRP levels. We grouped these variables into four categories: genetic, cell composition, intrinsic (age and sex) and exposures (smoking, CMV infection and CRP levels). For the $p$:th CpG site and the $i$:th individual, we collected observations of the minor allele count for the most associated local SNP in $x_i^{p,g}$, the proportions of the 15 cell types, excluding neutrophils, in the vector $\mathbf{x}_i^c$, intrinsic factors (sex and natural spline expanded values of age) in the vector $\mathbf{x}_i^{in}$ and exposures (smoking status, CMV serostatus and log-transformed CRP levels) in the vector $\mathbf{x}_i^e$, with corresponding parameters $\beta_g^p$, $\boldsymbol{\beta}_c^p$, $\boldsymbol{\beta}_{in}^p$ and $\boldsymbol{\beta}_e^p$. We interpret here log-transformed CRP levels as a proxy measure of the exposure of chronic low-grade inflammation. For each CpG site, we define linear predictor terms by

$$
f_g^p\left(x_i^{p,g}\right) = x_i^{p,g}\beta_g^p, \tag{19}
$$

$$
f_c^p\left(\mathbf{x}_i^c\right) = \left(\mathbf{x}_i^c\right)^t\boldsymbol{\beta}_c^p, \tag{20}
$$

$$
f_{in}^p\left(\mathbf{x}_i^{in}\right) = \left(\mathbf{x}_i^{in}\right)^t\boldsymbol{\beta}_{in}^p, \tag{21}
$$

$$
f_e^p\left(\mathbf{x}_i^e\right) = \left(\mathbf{x}_i^e\right)^t\boldsymbol{\beta}_e^p \tag{22}
$$

These functions vary in their degrees of freedom, so to get a fair comparison between them, we estimated group effect sizes as the out-of-sample proportion of variance explained by each group predictor. This estimation is done by indexing samples into two disjoint index groups $I_1$ and $I_2$, fitting the models on samples from $I_1$, and evaluating the prediction accuracy on samples from $I_2$.

Let $y_i^p$ be 5mC levels for the $p$:th CpG site on the β value scale. Take cell composition as example. To compute the total effect of cell composition on 5mC levels at the CpG site, we first fit a model with individuals in $I_1$,

$$
y_i^{p,c} = \mu^p + \left(\mathbf{x}_i^c\right)^t\boldsymbol{\beta}_c^p, i \in I_1 \tag{23}
$$

with parameters $\hat{\beta}_c^p$ and $\hat{\mu}^p$ estimated by least squares. We then define the total effect size to be the squared correlation between the observations and the out-of-sample predictions in individuals in $I_2$,

$$
\left(R_c^{\text{Tot}}\right)^2 = \text{cor}\left(y_j, \hat{y}_j^{p,c}\right)^2, j \in I_2. \tag{24}
$$

Total effects for the other predictor groups were defined analogously.

For groups other than the cell composition group, we also computed a direct effect. For each group, it was computed as the added out-of-sample proportion of variance explained when adding the group predictor term to that of the cell composition group. Take the exposures group as an example, the direct effect was computed by

$$
\left(R_e^D\right)^2 = \left(R_{e+c}^{\text{Tot}}\right)^2 - \left(R_c^{\text{Tot}}\right)^2, \tag{25}
$$

where $\left(R_{e+c}^{\text{Tot}}\right)^2$ is the total effect of the sum of the predictor terms for exposures and cell composition,

$$
f_{c+e} = f_c^p\left(\mathbf{x}_i^c\right) + f_e^p\left(\mathbf{x}_i^e\right). \tag{26}
$$

To mitigate the impact of sampling on estimates of total and direct effects, we did four independent repeats of five-fold cross-validation and averaged effect sizes across all 20 samples. To have an unbiased estimation of the out-of-sample explained variance, we redid a local meQTL mapping on the training set in each iteration of the cross-validation scheme. The algorithm for drawing samples of the total effect is detailed in Algorithm (2).

**Algorithm 2**. Cross-validation for estimating out-of-sample group total effect size.

Repeat 4 times:
With equal probability, assign an integer between 1 and 5 to all individuals.
For $k = 1,\ldots,5$
Index individuals assigned $k$ as $I_k$, the others are indexed as $I_{\backslash k}$
Select SNP for the predictor $f_g^p$ by performing a local meQTL mapping on individuals in $I_{\backslash k}$
For predictor $f_n^p \in \{f_g^p, f_c^p, f_{in}^p, f_e^p\}$
Estimate $\hat{\mu}^p$, $\hat{\beta}_n^p$ with $I_1 = I_{\backslash k}$
Compute $\left(R_n^{\text{Tot}}\right)^2$ by Eq. (24) with $I_2 = I_k$

The scheme to sample the direct effects is analogous. Finally, we estimated an effect size for interactions between the local SNP and non-genetic factors for each CpG site. It was computed, similarly to Eq. (25), as the added out-of-sample proportion of variance explained by the regression function,

$$
\begin{aligned}
f_{\text{Int}}^p\left(\text{SNP}_i^p, \text{Age}_i, \text{Woman}_i, \text{CMV}_i, \text{ExSmoker}_i, \text{Smoker}_i, \text{CRP}_i\right) \\
= \mu^p + \text{SNP}_i^p\beta_{\text{SNP}}^p + \text{Age}_i\beta_{\text{Age}}^p + \text{Woman}_i\beta_{\text{Woman}}^p + \text{CMV}_i\beta_{\text{CMV}}^p \\
+ \text{ExSmoker}_i\beta_{\text{ExSmoker}}^p + \text{Smoker}_i\beta_{\text{Smoker}}^p + \log(\text{CRP}_i)\beta_{\text{CRP}}^p \\
+ \text{SNP}_i^p\big(\text{Age}_i\theta_{\text{Age}}^p + \text{Woman}_i\theta_{\text{Woman}}^p + \text{CMV}_i\theta_{\text{CMV}}^p + \text{ExSmoker}_i\theta_{\text{ExSmoker}}^p \\
+ \text{Smoker}_i\theta_{\text{Smoker}}^p + \log(\text{CRP}_i)\theta_{\text{CRP}}^p\big)
\end{aligned}
$$
(27)

compared to the same regression function without interaction terms,

$$
\begin{aligned}
f_{\text{Main}}^p\left(\text{SNP}_i^p, \text{Age}_i, \text{Woman}_i, \text{CMV}_i, \text{ExSmoker}_i, \text{Smoker}_i, \text{CRP}_i\right) \\
= \mu^p + \text{SNP}_i^p\beta_{\text{SNP}}^p + \text{Age}_i\beta_{\text{Age}}^p + \text{Woman}_i\beta_{\text{Woman}}^p + \text{CMV}_i\beta_{\text{CMV}}^p \\
+ \text{ExSmoker}_i\beta_{\text{ExSmoker}}^p + \text{Smoker}_i\beta_{\text{Smoker}}^p + \log(\text{CRP}_i)\beta_{\text{CRP}}^p.
\end{aligned}
$$
(28)

## Biological annotations

Information about the position, closest gene and CpG density of each CpG site was obtained from the Illumina EPIC array manifest v.1.0 B4. We retrieved the chromatin state of regions around each CpG site, using the 15 chromatin states inferred with ChromHMM for CD4+ naive T cells by the ROADMAP Epigenomics consortium[15]. We used

peripheral blood mononuclear cells (PBMCs) as reference. The data was downloaded from the consortium webpage (https://egg2.wustl.edu/roadmap/web_portal/chr_state_learning.html). The transcription factor binding site data used was public CHIP-seq data collected and processed for the 2020 release of the ReMap database[118], including a total of 1165 TFs. Binding sites include both direct and indirect binding. Enrichment analyses were performed by creating simple two-way tables for each target set and each annotation (i.e., chromatin states, CpG density, transcription factor binding site), and then performing Fisher's exact test. Gene ontology enrichments were computed with the gometh function in the missMethyl R package[119].

We tested if a set of $x$ local or remote meQTL SNPs is enriched in disease- or trait-associated variants, by sampling at random, among all tested SNPs, 15,000 sets of $x$ SNPs with minor allele frequencies matched to those of meQTL SNPs. For each resampled set, we calculated the proportion of variants either known to be associated with a disease or trait, or in LD (set here to $r^2 > 0.6$) with a disease/trait-associated variant ($P$-value $< 5 \times 10^{-8}$; EBI-NHGRI Catalog of GWAS hits version e100 r2021-01-1). The enrichment $P$-value was estimated as the percentage of resamples for which this proportion was larger than that observed in meQTL SNPs. LD was precomputed for all 5,699,237 SNPs with PLINK 1.9 (with arguments '–show-tags all–tag-kb 500–tag-r2 0.6')[120].

### Reporting summary
Further information on research design is available in the Nature Research Reporting Summary linked to this article.

### Data availability
The Infinium MethylationEPIC raw and processed data generated in this study[27] have been deposited in the Institut Pasteur data repository, OWEY, which can be accessed via the following link: https://doi.org/10.48802/owey.f83a-1042. All association statistics obtained in this study (i.e., the 141 EWAS and interaction models, local meQTL mapping) can be explored and downloaded from the web browser http://mimeth.pasteur.fr/. The SNP array data can be accessed in the European Genome-Phenome Archive (EGA) with the accession code EGAS00001002460. All Milieu Intérieur datasets can be accessed by submitting a data access request to milieuinterieurdac@pasteur.fr, the Milieu Intérieur data access committee (DAC). The DAC informs all the research participants of the data access request and grants data access if the request is consistent with the informed consent signed by the participants. In particular, research on Milieu Intérieur datasets is restricted to research on the genetic and environmental determinants of human variation in immune responses. Data access is typically granted two months after request submission.

### Code availability
All the code supporting the current study, including the CMV estimation model, has been uploaded to GitHub:[121] https://github.com/JacobBergstedt/MIMETH.

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

## Acknowledgements

We thank Sarah Merrill, Nicole Gladish, Violaine Saint-André, Lucas Husquin and the Milieu Intérieur scientific advisory board for comments and helpful discussions. We acknowledge the help of the HPC Core Facility of Institut Pasteur for this work. This research was enabled, in part, by the use of the FlowSorted.BloodExtended.EPIC R package developed at Dartmouth College, which software is subject to the licensing terms made available by Dartmouth Technology Transfer and which software is provided "as is" with no warranties whatsoever. This work benefited from support of the French government's program 'Investissement d'Avenir', managed by the Agence Nationale de la Recherche (reference 10-LABX-69-01).

## Author contributions

L.Q.-M. initiated the study. J.B., E.P., and L.Q.-M. conceived and developed the study. A.U. prepared DNA samples. D.T.S.L., J.L.M., and M.S.K. acquired Illumina Infinium MethylationEPIC array data. J.B. performed all analyses, with contributions from S.A.K.A., K.T., and E.P. E.P. supervised all analyses. A.J. developed the web browser. D.D. and M.L.A. advised on experiments. M.R., M.S.K., D.D., and M.L.A. advised on data interpretation. J.B., E.P., and L.Q.-M. wrote the manuscript. All authors discussed the results and contributed to the final manuscript.

## Competing interests

The authors declare no competing interests.

## Additional information

## Milieu Intérieur Consortium

Laurent Abel[8], Andres Alcover[9], Hugues Aschard[10], Philippe Bousso[11], Nollaig Bourke[12], Petter Brodin[13,14], Pierre Bruhns[15], Nadine Cerf-Bensussan[16], Ana Cumano[17], Christophe d'Enfert[11], Ludovic Deriano[18], Marie-Agnès Dillies[19], James Di Santo[20], Françoise Dromer[21], Gérard Eberl[22], Jost Enninga[23], Jacques Fellay[24,25,26], Ivo Gomperts-Boneca[27], Milena Hasan[28], Gunilla Karlsson Hedestam[29], Serge Hercberg[30], Molly A. Ingersoll[31], Olivier Lantz[32,33], Rose Anne Kenny[34,35], Mickaël Ménager[36], Frédérique Michel[37], Hugo Mouquet[38], Cliona O'Farrelly[39,40], Etienne Patin ®[1,53] ✉, Sandra Pellegrini[37], Antonio Rausell[41], Frédéric Rieux-Laucat[42], Lars Rogge[43], Magnus Fontes[44], Anavaj Sakuntabhai[45], Olivier Schwartz[46], Benno Schwikowski[47], Spencer Shorte[48], Frédéric Tangy[49], Antoine Toubert[50], Mathilde Touvier[30], Marie-Noëlle Ungeheuer[51], Christophe Zimmer[52], Matthew L. Albert ®[4], Darragh Duffy ®[6] & Lluís Quintana-Murci[1,7]

[8]Imagine Institute, University Paris Cité, Necker Hospital for Sick Children, INSERM UMR 1163, Laboratory of Human Genetics of Infectious Diseases, Paris, France. [9]Institut Pasteur, Université Paris Cité, INSERM-U1224, Unité Biologie Cellulaire des Lymphocytes, Ligue Nationale Contre le Cancer, Équipe Labellisée Ligue, 2018 Paris, France. [10]Institut Pasteur, Université Paris Cité, Department of Computational Biology, Paris, France. [11]Institut Pasteur, Université Paris Cité, INRAE USC2019, Unité Biologie et Pathogénicité Fongiques, Paris, France. [12]Department of Medical Gerontology, School of Medicine, Trinity College Dublin, Dublin, Ireland. [13]Department of Immunology and Inflammation, Imperial College London, London, UK. [14]Department of Women's and Children's Health, Karolinska Institutet, Stockholm, Sweden. [15]Institut Pasteur, Université Paris Cité, INSERM UMR1222, Unit of Antibodies in Therapy and Pathology, Paris, France. [16]Institut Imagine, Université Paris Cité, INSERM UMR1163, Laboratory Intestinal Immunity, Paris, France. [17]Institut Pasteur, Université

Paris Cité, INSERM U1223, Unit Lymphocytes and Immunity, Paris, France. [18]Institut Pasteur, Université Paris Cité, INSERM U1223, Équipe Labellisée Ligue Contre Le Cancer, Genome Integrity, Immunity and Cancer Unit, Paris, France. [19]Institut Pasteur, Université Paris Cité, Bioinformatics and Biostatistics Hub, Paris, France. [20]Institut Pasteur, Université Paris Cité, INSERM U1223, Innate Immunity Unit, Paris, France. [21]Institut Pasteur, Université Paris Cité, CNRS UMR2000, Unité de Mycologie Moléculaire, Centre national de Référence Mycoses Invasives et Antifongiques, Paris, France. [22]Institut Pasteur, Université Paris Cité, INSERM U1224, Microenvironment and Immunity Unit, Paris, France. [23]Institut Pasteur, Université Paris Cité, CNRS UMR3691, Dynamics of Host-Pathogen Interactions Unit, Paris, France. [24]School of Life Sciences, École Polytechnique Fédérale de Lausanne, Lausanne, Switzerland. [25]Swiss Institute of Bioinformatics, Lausanne, Switzerland. [26]Precision Medicine Unit, Lausanne University Hospital and University of Lausanne, Lausanne, Switzerland. [27]Institut Pasteur, Université Paris Cité, CNRS UMR2001, Unité Biologie et Génétique de la Paroi Bactérienne, Paris, France. [28]Institut Pasteur, Université Paris Cité, Cytometry and Biomarkers Unit of Technology and Service, Paris, France. [29]Department of Microbiology, Tumor and Cell Biology, Karolinska Institutet, Stockholm, Sweden. [30]Université Sorbonne Paris Nord, Université de Paris, INSERM U1153, INRAE U1125, CNAM, Epidemiology and Statistics Research Center, Nutritional Epidemiology Research Team, Bobigny, France. [31]Institut Pasteur, Université Paris Cité, Institut Cochin, INSERM U1016, CNRS UMR 8104, Mucosal Inflammation and Immunity Group, Paris, France. [32]Institut Curie, Université Paris Sciences et Lettres, INSERM U932, Laboratoire d'Immunologie Clinique, Paris, France. [33]Centre d'Investigation Clinique en Biothérapie Gustave-Roussy Institut Curie (CIC-BT1428), Paris, France. [34]The Irish Longitudinal Study on Ageing (TILDA), Trinity College Dublin, Dublin, Ireland. [35]Mercer's Institute for Successful Ageing, St James's Hospital, Dublin, Ireland. [36]Imagine Institute, Université Paris Cité, INSERM UMR1163, Laboratory of Inflammatory Responses and Transcriptomic Networks in Diseases, Atip-Avenir Team, Paris, France. [37]Institut Pasteur, Université Paris Cité, INSERM U1221, Cytokine Signaling Unit, Paris, France. [38]Institut Pasteur, Université Paris Cité, INSERM U1222, Laboratory of Humoral Immunology, 75015 Paris, France. [39]Comparative Immunology, School of Biochemistry and Immunology, Trinity Biomedical Sciences Institute, Dublin, Ireland. [40]School of Medicine, Trinity College Dublin, Dublin, Ireland. [41]Imagine Institute, Université Paris Cité, INSERM UMR1163, Necker Hospital for Sick Children, Clinical Bioinformatics Laboratory, Paris, France. [42]Imagine Institute, Université Paris Cité, INSERM UMR 1163, Laboratory of Immunogenetics of Autoimmune Diseases in Children, Paris, France. [43]Institut Pasteur, Université Paris Cité, AP-HP Hôpital Cochin, Immunoregulation Unit, Paris, France. [44]Institut Roche, Boulogne-Billancourt, France. [45]Institut Pasteur, Université Paris Cité, CNRS UMR2000, Functional Genetics of Infectious Diseases Unit, Paris, France. [46]Institut Pasteur, Université Paris Cité, CNRS UMR3569, Virus and Immunity Unit, Paris, France. [47]Institut Pasteur, Université Paris Cité, Computational Systems Biomedicine Lab, Paris, France. [48]Institut Pasteur, Université Paris Cité, UTechS-PBI/Imagopole, Paris, France. [49]Institut Pasteur, Université Paris Cité, CNRS UMR3965, Viral Genomics and Vaccination Unit, Paris, France. [50]AP-HP, Hôpital Saint-Louis, Université de Paris, INSERM UMR1160, Laboratoire d'Immunologie et d'Histocompatibilité, Paris, France. [51]Institut Pasteur, Université Paris Cité, Investigation Clinique et Accès aux Ressources Biologiques (ICAReB), Paris, France. [52]Institut Pasteur, Université Paris Cité, CNRS UMR3691, Imaging and Modeling Unit, Paris, France.

