## [Peer Review File · Nature Communications]

The Immune Factors Driving DNA Methylation Variation in Human BloodREVIEWER COMMENTS

Reviewer #1 (Remarks to the Author):

The paper "Factors Driving DNA Methylation Variation in Human Blood" presents a general analysis of factors driving DNA methylation variation in a cohort of approximately 1000 individuals (Western European) with matched SNP profiling and deep immunophenotyping as well as other epidemiological factors. The authors aim to compare various factors (genetic, cell-type composition, exposures) in their contribution to DNAm variation, although in parallel to this there are also many in-depth analysis of mQTLs, age and smoking effects and DNAm changes associated with viral infections.

In my opinion, although I did not find this manuscript to be conceptually novel, it does have in principle two major "selling points":

(1) It presents a valuable resource, a novel DNA methylation dataset with matched genotype and cell-composition information, which however is not made readily available since deposition to EGA requires users to submit detailed requests for data-access, which is not in the spirit of reproducibility and not aligned with the principles of open data access, since the authors of this study can always refuse data access. I personally have very little sympathy for authors who engage in this practice, since the DNA methylation, cell-composition and epidemiological data could (and should) be made available on a more user-friendly repository such as GEO. The genotype data is understandably to be placed on EGA. Hence, for me deposition of the DNAm+cell-composition+epidemiological data to GEO is one condition for this paper to be acceptable for publication in a high impact journal like Nat Commun, because the work needs to be reproducible and the value of this paper as a RESOURCE needs to be realized through unconditional deposition of data in a public repository without the need for specific data-access requests.

(2) The study makes an interesting observation that latent cytomegalovirus infection is a major driver of DNA methylation variation, accounting for more variation in DNAm than factors such as age and smoking, and although this is mentioned in the abstract, it does not actually feature prominently within the Results section because this paper covers far too much ground which is actually not novel.

Indeed, I am sorry to say but many of the results reported in this manuscript, specially those related to aging, smoking and mQTLs are not novel at all and yet the authors discuss these results as if they were entirely novel. In doing so, the manuscript becomes unreasonably long and I personally got bored of reading it half-way through because I could not spot anything really novel or exciting until the very last sections on the CMV and interaction analyses. Some of the insights on the mQTL analysis (the ZFP analysis) is undoubtedly interesting but how does it really fit within the title implied by this MS? Indeed, the title of the manuscript is also a little boring in my opinion, as the EWAS community already knows that cell-type composition and genetic variation are the dominant factors, so in this regard there is nothing in this manuscript that can be considered "ground-breaking". Personally, I also think that this manuscript tries to cover far too many topics (e.g. mQTLs, aging, viral exposures, smoking, BMI, cell-type specific analysis, genotype environment interactions), so that some of the more interesting findings are effectively "lost" in a large sea of known findings. So, I feel that this manuscript needs more focus and that it needs to be completely rewritten. I would advise that if the CMV-infection is such a major contributor of DNAm variation in healthy cohorts, that the authors should provide a more in-depth analysis of what the actual consequence of this is for the interpretation of EWAS data. Is CMV a confounder for other factors? What happens if we don't adjust for it when performing EWAS for other factors such as age, smoking, BMI etc? Does it affect results or not? The authors do not provide any indication as to the real significance and implications of this DNAm variation, and yet the reader is left wanting to know more about it....

Besides this major structural and organizational flaw, I also have some other major concerns which further dampen enthusiasm for this manuscript:

Other major concerns:

1) Lack of novelty: A lot of the results reported in this MS, for instance those related to aging and

smoking are not novel at all and have already been noted by previous studies. For instance, the increased dispersion of DNAm with age was already noted in Slieker RC et al Genome Biol.2016. That most age-associated changes are not dependent on cell-type was noted in Zhu T et al Aging 2018.

2) Most EWAS studies have explored whether DNAm changes are driven or not by shifts in cell-type composition, albeit not at the cellular resolution of 16 cell-types as done here. For this reason, I would suggest that the authors focus their discussion a lot more on the impact of doing cell-type correction with 16 cell-types as opposed to the usual 6-7. This is done in a few places, but this is an important insight that needs a lot more emphasizing, so much so that it would also deserve a separate paper. As currently written, the importance of cell-type correction at higher resolution is an important message that however does not feature prominently enough.

3) Related to the previous point, it would be important to determine that results are robust to the choice of DNA methylation reference matrix. My understanding is that the authors have used the one by Koestler et al, but there are others like the one originally used by Houseman. It would also be important to test robustness to the actual statistical algorithm used, as sometimes there can be discrepancies. In addition, the authors should provide boxplots displaying the estimated cell-type fractions for all 16 cell subtypes in their cohort.

4) Line 302: the analysis described here is a critical component of this manuscript. As described, it would appear that the authors treated each factor (e.g. CMV infection, smoking, BMI) separately, which leaves open the possibility of confounding between factors themselves. In other words, from the analysis done by the authors we don't know if the surprisingly large number of CpGs associated with CMV infection is not due to other factors that are themselves associated with CMV infection. Indeed, it would be good if the authors could list with epidemiological factors (including age) are associated with CMV infection, and if so what the impact would be if the authors were to combine factors in the same model.

5) Related to the previous point, I would like to also see a complementary unsupervised analysis to back up the claims made by the authors in relation to CMV being a major driver of DNAm variation. For instance, if the authors were to do a SVD/PCA, extract all significant components of variation, and then correlate each of these components to the major factors in a multivariate framework that accounts for the associations between these factors, this would then be in my opinion a much more convincing way to demonstrate the impact of CMV infection on DNAm variation. The authors should also note that supervised analyses are much more sensitive to unaccounted confounders, specially when using the FDR-metric....which also has a large variance anyway. Bonferroni-adjustment provides a less sensitive metric which is more appropriate in the supervised-analysis context that the authors have pursued in their MS.

6) Cell-type specific analyses: I understand that the authors use an interaction model, but I could not appreciate what is different in the mathematical exposition presented in this MS from the CellDMC or TOAST methods published by other groups. Are the authors justified to include all these mathematical equations if they are not actually novel? A related point is that when discussing the results of the cell-type specific analyses, they always report them "relative" to the neutrophil fraction, which I find rather confusing. Given that the neutrophil fraction is highest in blood, it is extremely important if the authors could report the number of cell-type specific changes in the neutrophil component in addition to the other cell-types. It should be noted that even with 1000 samples the authors could be seriously underpowered to detect changes in as many as 6 cell-types. One should also be careful in noting that the power to detect changes in a particular cell-type will depend on the variation in the fraction displayed by that cell-type, so this is variable between cell-types, which can make these type of analyses quite problematic. In view of this, the notion of cell-type specific needs to be clarified, since a reader may interpret this as changing in one of the 6 cell-type but not in the other 5, whilst someone else might interpret this as a statement of certainty that the change is happening in a given cell-type without necessarily implying how it changes in the other cell-types if at all. More clarification by the authors is required.

Reviewer #2 (Remarks to the Author):

Bergstedt and colleagues report on a comprehensive analysis of factors associated with DNA methylation differences in blood. The cohort is beautiful with a large age range, equal numbers of men and women, a wealth of molecular and epidemiological data, and, crucially, detailed information on cell composition of the blood samples and CMV status is present. Also, the paper is well-written, and analyses generally seem to be well-thought through. The work has substantial potential to be of great interest to the population epigenetics community. While I started out reading the manuscript with great expectation, I was somewhat disappointed when I finished it. On the one hand I read many things I already knew from previous work (while science is about novelty), on the other hand I missed more depth regarding the analysis of aspects that are novel (cell counts and CMV effects, relative contribution of factors). A shift in focus to novelty would improve the paper, in particular when novel aspects were analyzed in detail than done now in a manuscript that deals with almost any aspect related to the field.

- Repeatedly analysis and results are not explicitly described as a confirmation of previous work, work that often was more in depth than in the current manuscript that covers so many aspects. Readers may be left with the impression that the authors think results are new. When confirming previous results, a direct comparison of results is required (to what extent were the same CpGs found?) and a description of the new findings in addition to the previous work. Obviously, it should be made specific which previous work did the novel observation. That said: it is important for scientific progress to confirm previous findings. A non-complete list of examples includes:

+ The manuscript presents as main finding (start discussion) a role of ZNF transcription factors, a role of KRAB, and SENP7 as example. SENP7 was originally found by Lemire et al (Nat Commun 2015). The result regarding ZNF transcription factors was a discovery reported by Hop et al (Genome Biol 2020). The discussion of the current manuscript reads as virtually identical to the latter paper (KRAB, deSUMOylation of KAP1). Both papers were already cited by the authors.

+ The characterization of meQTLs contains little news. There now are so many meQTL papers (including recently a meta-analysis of 30k samples; Min et al Nat Genet 2021). It remains unclear what new observations the authors made if any.

+ Epigenetic dispersion has been studied at a genome-wide level in detail previously. The authors may have missed the work of Zhang et al (Genome Med 2018) and Slieker et al (Genome Biol 2016). It will be informative to compare the CpGs common between the EPIC and 450k array. What is different? What insights are new? The interpretation of findings is very similar to the earlier reports, which is reassuring (and should be made explicit).

+ I enjoyed reading details like on DYRK2 promotor methylation in relation to sex-specific changes. It would be important to know if this is the first time that these pieces of information are linked, or this was already done before.

- Various aspects of the statistical analysis require more detail. Models fitted are complex with many (correlated) variables entered together, which requires reporting diagnostics to be sure nothing went wrong in fitting the models. Also, comparisons of analysis with and without cell counts are only done at the level of genome-wide significant p-values, while a more explicit testing of hypotheses would lead to stronger inference.

+ Cell counts add up to 100%. This implies that models may not fit well when all counts are entered (e.g. issues with model intercept SE). Also, cell counts can be (strongly) correlated leading to collinearity. The authors should provide a heat map of correlations between their measured cell counts (perhaps combined with IDOL-predicted cell counts). Also, they need to address co-linearity and the fact that including all counts resulting in at least one parameter not being estimable.

- + The authors need to report (and if necessary correct for) inflation of test statistics. In particular in combination with FDR, this may have large effects.
- + The authors report the number of FDR-significant CpGs in a model without cell counts, with cell counts and those that are no longer $FDR < 0.05$ with cell counts in the model. This provides a superficial and incomplete picture because very-low-p-value criteria are highly influenced by power instead of biology (slight change may render effect no longer significant).
- + By comparing FDR significant findings from separate analyses, the authors cannot distinguish two effects of including cell counts: a mediator and noise-reduction (leading to more power).
- + While, the authors interpret their findings as mediation, this is not tested. The authors should explicitly perform mediation tests before they claim mediation (e.g. for CpGs with a 'total effect' only). Addition of mediation analysis is key in my opinion when improving the work.
- + The cell types measured are of exceptional interest going from naïve to terminally differentiated cells. I was very eager to read more on the contribution of the various cell types. Simply entering all cell types provides little insight in the actual drivers of DNA methylation variation. We already know that cell counts matter a lot. The new data this study may bring is which specific cell types are important. More focus on this question (in combination with mediation analysis) would be of great interest to the field.
- + Similar reasoning is valid for CMV: the observation that CMV matters (and not other viruses) is highly relevant, but is overshadowed in the current manuscript by findings repeating previous work, while more in-depth analyses regarding CMV would be of great interest.
- Is there evidence of dispersion in cell counts with age? Important to report.
- Smoking-associated differential DNA methylation at GRP15 was suggested to be induced by T cell composition changes (Bauer et al. Clin Epigenet 2015). Is this something the authors could confirm or exclude (in relation to the conclusion that smoking-associated methylation differences are often not related to cell counts).
- Heparin negatively affects DNA studies. How did the authors address this problem?
- The authors excluded 118k probes overlapping with any SNP $MAF > 1\%$. Was this irrespective of the exact location of the SNP relative to the probe and, if so, why?
- Data are not yet available from EGA? The authors are to be commended for creating a browser that enables others to download summary stats.
- Why did the authors not for example use PBMC chromatin states? Choice for a single cell type is unclear.

Responses to reviewers

Reviewer #1 (Remarks to the Author):

The paper “Factors Driving DNA Methylation Variation in Human Blood” presents a general analysis of factors driving DNA methylation variation in a cohort of approximately 1000 individuals (Western European) with matched SNP profiling and deep immunophenotyping as well as other epidemiological factors. The authors aim to compare various factors (genetic, cell-type composition, exposures) in their contribution to DNAm variation, although in parallel to this there are also many in-depth analysis of mQTLs, age and smoking effects and DNAm changes associated with viral infections.

We thank the reviewer for their thoughtful comments. We would like to clarify that, in the initial version of the manuscript, our objective was not to present known findings as if they were new, but to confirm several previous findings while correcting for cellular heterogeneity more completely than previous studies. Furthermore, one of our main goals was to provide a quantitative assessment of the respective effects of various factors known to affect DNAm, in the same study and cohort, so that they can be directly compared. Nevertheless, we understand that this can be perceived as less interesting than the newly described effects of CMV on DNAm variation and the comparison of corrections for cellular heterogeneity. Consequently, we have now extensively modified our manuscript, to give more emphasis on the most novel findings, in line with the reviewer’s comments. Namely, we now highlight our results on (i) the effects of CMV infection on DNAm variation, including new analyses to evaluate its role as a possible mediator in EWAS and a new algorithm to predict CMV serostatus from DNA methylation data, and (ii) the comparison of corrections on cellular heterogeneity based on measured or estimated blood cell subsets.

We would like to note that, when revising the manuscript, we realized that our results were more robust when excluding 74 donors for which a large fraction of the flow cytometric data was missing. These missing values were initially imputed, but because imputation was not totally accurate, it introduced noise in the statistical inferences. All analyses are now done on 884 donors who have both accurate DNAm and flow cytometric data.

All revisions are highlighted *in gray* in the revised version of the manuscript.

In my opinion, although I did not find this manuscript to be conceptually novel, it does have in principle two major “selling points”:

(1) It presents a valuable resource, a novel DNA methylation dataset with matched genotype and cell-composition information, which however is not made readily available since deposition to EGA requires users to submit detailed requests for data-access, which is not in the spirit of reproducibility and not aligned with the principles of open data access, since the authors of this study can always refuse data access. I personally have very little sympathy for authors who engage in this practice, since the DNA methylation, cell-composition and epidemiological data could (and should) be made available on a more user-friendly repository such as GEO. The genotype data is understandably to be placed on EGA. Hence, for me deposition of the DNAm+cell-composition+epidemiological data to GEO is one condition for this paper to be acceptable for publication in a high impact journal like Nat Commun, because the work needs to be reproducible and the value of this paper as a RESOURCE needs to be realized through unconditional deposition of data in a public repository without the need for specific data-access requests.

Our objective is indeed to provide a useful resource to academic research, which is why we developed a web browser that enables direct access to all the statistical results of our study

(<http://mimeth.pasteur.fr/>). We completely agree with the reviewer that we should try to facilitate reproducibility and promote open science as much as possible, by providing direct access to our DNAm data. Nevertheless, we are restricted in these objectives by national laws regarding French research participants. According to these laws, it is compulsory that the Milieu Intérieur donors are aware of any research project that utilizes their data. Namely, we are asked to list on the Milieu Intérieur website (link) any project, from the Consortium or any other research lab, that uses Milieu Intérieur datasets. Consequently, we will make the raw and processed DNAm data fully available, but data requesters will have to submit a (short) data access request to the Milieu Intérieur data access committee (DAC), in which they briefly describe what they plan to do with the data. So far, our DAC has accepted and openly shared data for nearly all received requests. The data will be made available upon manuscript publication, through our new institutional data repository, OWEY, recently developed by Institut Pasteur, which will avoid previous challenges and delays that we have experienced with EGA.

(2) The study makes an interesting observation that latent cytomegalovirus infection is a major driver of DNA methylation variation, accounting for more variation in DNAm than factors such as age and smoking, and although this is mentioned in the abstract, it does not actually feature prominently within the Results section because this paper covers far too much ground which is actually not novel.

Indeed, I am sorry to say but many of the results reported in this manuscript, specially those related to aging, smoking and mQTLs are not novel at all and yet the authors discuss these results as if they were entirely novel. In doing so, the manuscript becomes unreasonably long and I personally got bored of reading it half-way through because I could not spot anything really novel or exciting until the very last sections on the CMV and interaction analyses. Some of the insights on the mQTL analysis (the ZFP analysis) is undoubtedly interesting but how does it really fit within the title implied by this MS? Indeed, the title of the manuscript is also a little boring in my opinion, as the EWAS community already knows that cell-type composition and genetic variation are the dominant factors, so in this regard there is nothing in this manuscript that can be considered “ground-breaking”. Personally, I also think that this manuscript tries to cover far too many topics (e.g. mQTLs, aging, viral exposures, smoking, BMI, cell-type specific analysis, genotype environment interactions), so that some of the more interesting findings are effectively “lost” in a large sea of known findings. So, I feel that this manuscript needs more focus and that it needs to be completely rewritten. I would advise that if the CMV-infection is such a major contributor of DNAm variation in healthy cohorts, that the authors should provide a more in-depth analysis of what the actual consequence of this is for the interpretation of EWAS data. Is CMV a confounder for other factors? What happens if we don’t adjust for it when performing EWAS for other factors such as age, smoking, BMI etc? Does it affect results or not? The authors do not provide any indication as to the real significance and implications of this DNAm variation, and yet the reader is left wanting to know more about it...

We thank the reviewer for this thorough comment and advice. Following the reviewer’s suggestions, we have now: (i) added several new results that describe the effects of CMV on 5mC levels in human blood, including EWAS that are not adjusted on CMV and associations between CMV and principal components of DNAm levels; (ii) reduced substantially (or moved to supplementary text) the confirmatory results, including the effects of genotypes, smoking and CRP levels on DNAm; and (iii) better justified why we revisit the effects of age on DNAm variation, as age effects can be mediated by CMV infection or proportions of cell subsets that were not accounted for in previous studies. Furthermore, we have also developed a predictive model of CMV serostatus from DNAm data, to enable future EWAS to control for confounding by CMV.

Our new results show that, when not adjusting for cell proportions, or when adjusting on cells estimated by IDOL, CMV serostatus is strongly associated with PC1 and PC2 of the DNA methylation data, whereas age and sex explain PC3 to PC5. However, when adjusting on 16 measured cell proportions, PC1 and PC2 are no longer associated with CMV, which are actually explained by immune cell proportions, including CD4⁺ and CD8⁺ memory T cells. These results imply that most CMV effects on DNAm are mediated by blood cell composition, in line with our initial conclusions, and the fact that CMV infection causes a strong increase in memory T cell subsets (Patin*, Hasan*, Bergstedt* et al., *Nat Immunol* 2018). Importantly, naïve and memory T cell fractions are not estimated by standard cell mixture deconvolution methods, which suggests that CMV can affect EWAS results when these methods are used.

We also found statistical evidence at 10,074 CpG sites that CMV mediates total effects of age on DNAm. Accordingly, age has a significant total effect on 5mC levels at 97,219 and 113,742 CpG sites when adjusting or not on CMV serostatus, which is why we revisited the well-described age effects on DNAm, but adjusting on CMV. Expectedly, we found that CMV infection does not mediate the effects of smoking, sex, BMI or CRP levels as these factors are not associated with CMV serostatus. We anticipate that CMV may confound EWAS of diseases associated with CMV, such as cardiovascular disease (Wang et al., *J Am Heart Assoc* 2017).

All these new results are now included in the revised manuscript, lines 149-167 and 195-246.

Other major concerns:

1) Lack of novelty: A lot of the results reported in this MS, for instance those related to aging and smoking are not novel at all and have already been noted by previous studies. For instance, the increased dispersion of DNAm with age was already noted in Slieker RC et al Genome Biol.2016. That most age-associated changes are not dependent on cell-type was noted in Zhu T et al Aging 2018.

In addition to the major changes described above, we have now verified that our results confirm previous findings and have cited systematically relevant studies. In particular, we have compared the CpG sites that are the most dispersed with age with those found by a previous study (Slieker et al Genome Biol 2016). Reassuringly, we replicated 2,604 out of 5,075 CpG sites previously reported to be increasingly variable with age, supporting a strong overlap between the two different approaches (OR 95% CI: [36.2, 40.8]). Importantly, our model is expected to account for blood cell composition more completely than previous models, as it includes measured levels of memory T cells, which show a strong increase in variance with age (see new Fig. 3E). Furthermore, our model tests whether DNAm *changes* with age, which enabled us to determine if DNAm variance typically *increases* or *decreases* with age, contrarily to previous approaches. Strikingly, we found that 99.4% of CpGs with age-related dispersion show an increase in the variance of 5mC levels with age, in line with an age-related decrease in the fidelity of the epigenetic maintenance machinery. These results are further described in lines 280-303.

2) Most EWAS studies have explored whether DNAm changes are driven or not by shifts in cell-type composition, albeit not at the cellular resolution of 16 cell-types as done here. For this reason, I would suggest that the authors focus their discussion a lot more on the impact of doing cell-type correction with 16 cell-types as opposed to the usual 6-7. This is done in a few places, but this is an important insight that needs a lot more emphasizing, so much so that it would also deserve a separate paper. As currently written, the importance of cell-type correction at higher resolution is an important message that however does not feature prominently enough.

Following the reviewer's comments, we have now included a new Results section relating to the performance of different corrections for cellular heterogeneity. We now compare EWAS results corrected for 16 cell types measured by FACS or cell types estimated by Houseman *et al.*'s method, IDOL method or the recently published EPIC IDOL-Ext method. We found that EWAS adjusted by Houseman *et al.*'s and IDOL methods detected more CpG sites associated with most candidate factors, relative to EWAS adjusted on the measured proportions of 16 cell types, particularly for age ($n = 131,142$ vs. $35,701$) and CMV infection ($n = 31,159$ vs. 245). Conversely, EWAS adjusted by the EPIC IDOL-Ext method, which estimates subsets of naïve and memory CD4⁺ and CD8⁺ T cells (Salas *et al.*, Nat Commun 2022), provide results that are similar to those of EWAS adjusted for high-resolution flow cytometric data. These results suggest that standard deconvolution methods only partially account for blood cellular heterogeneity, due to their low resolution. To further test this scenario, we have conducted EWAS adjusted on flow cytometric data for only six major cell types and found results comparable to those for Houseman *et al.*'s and IDOL methods. Together, these results indicate that standard adjustments for the proportions of only the six major cell-types are not able to fully account for blood cell heterogeneity, particularly when estimating the effects of age and CMV infection on DNA methylation, two factors that are known to skew CD4⁺ and CD8⁺ T cell compartments toward differentiated phenotypes (Patin, Hasan, Bergstedt *et al.*, Nat Immunol 2018). These new results can be found in lines 136-193. Additionally, we discuss more extensively these findings in lines 495-503.

3) Related to the previous point, it would be important to determine that results are robust to the choice of DNA methylation reference matrix. My understanding is that the authors have used the one by Koestler *et al.*, but there are others like the one originally used by Houseman. It would also be important to test robustness to the actual statistical algorithm used, as sometimes there can be discrepancies. In addition, the authors should provide boxplots displaying the estimated cell-type fractions for all 16 cell subtypes in their cohort.

We entirely agree with the reviewer. We now report EWAS results adjusted by either Houseman *et al.*'s, IDOL or EPIC IDOL-Ext methods. Notably, the first two deconvolution methods only partially account for blood cell composition when performing EWAS of age and CMV, probably because they do not estimate sub-types of CD4⁺ and CD8⁺ memory T cells, which are known to be strongly altered by age and CMV infection. Furthermore, we have now included plots showing measured vs. estimated cell proportions, for those that are in common between our flow cytometric data and reference-based deconvolution methods (see new Supplementary Fig 3).

We would like to clarify that we do not estimate cell-type fractions for the 16 cell subtypes, as they are measured directly from the donor's blood by flow cytometry (see Introduction, Results and Methods sections). This is a central novelty of our manuscript, which motivated us to revisit previous findings relating to age effects on DNAm, a factor that substantially alters blood cell composition (Patin, Hasan, Bergstedt *et al.*, Nat Immunol 2018). We now provide scatter plots of blood cell fractions measured by flow cytometry (see new Supplementary Fig 3).

4) Line 302: the analysis described here is a critical component of this manuscript. As described, it would appear that the authors treated each factor (e.g. CMV infection, smoking, BMI) separately, which leaves open the possibility of confounding between factors themselves. In other words, from the analysis done by the authors we don't know if the surprisingly large number of CpGs associated with CMV infection is not due to other factors that are themselves associated with CMV infection. Indeed, it would be good if the authors could list with epidemiological factors (including age) are associated with CMV infection, and if so what the impact would be if the authors were to combine factors in the same model.

We have now clarified further that *all* our EWAS models are systematically adjusted for age, sex, smoking and CMV serostatus, as well as the most associated mQTL (lines 141-145). As to BMI, given that CpG sites are associated with this factor, we initially thought that this factor was unlikely to affect results for the EWAS of CMV. Nevertheless, we have now conducted a new EWAS of CMV serostatus adjusting on BMI, as requested by the reviewer. We found that this has minimal effects on the results: CMV has a significant direct effect on 245 and 243 CpG when adjusting of not for BMI (data not shown).

5) Related to the previous point, I would like to also see a complementary unsupervised analysis to back up the claims made by the authors in relation to CMV being a major driver of DNAm variation. For instance, if the authors were to do a SVD/PCA, extract all signification components of variation, and then correlate each of these components to the major factors in a multivariate framework that accounts for the associations between these factors, this would then be in my opinion a much more convincing way to demonstrate the impact of CMV infection on DNAm variation.

We thank the reviewer for this interesting comment. As detailed above, we have now identified the factors that are associated with the first PCs of the DNAm data, by using a multiple regression model adjusted or not on blood cell composition. Our new results show that, when not adjusting for cell proportions, or when adjusting on cells estimated by IDOL, CMV serostatus is strongly associated with PC1 and PC2, whereas age and sex explain PC3 to PC5. However, when adjusting on 16 measured cell proportions, PC1 and PC2 are no longer associated with CMV, which are actually explained by immune cell proportions, including CD4⁺ and CD8⁺ memory T cells. These results imply that most CMV effects on DNAm are mediated by blood cell composition, in line with our initial conclusions. These results are now described in detail in lines 149-167. Importantly, although most CMV effects on DNAm are mediated by changes in immune cell proportions, our analyses also provide statistical evidence that CMV infection has a strong direct (cell-composition-independent) effect on DNAm, particularly at BRD4 binding sites, a host transcription factor that plays a key role in the regulation of CMV latency (Groves et al., *PNAS* 2021).

The authors should also note that supervised analyses are much more sensitive to unaccounted confounders, specially when using the FDR-metric....which also has a large variance anyway. Bonferroni-adjustment provides a less sensitive metric which is more appropriate in the supervised-analysis context that the authors have pursued in their MS.

Following the reviewer's suggestion, we now use the Bonferroni correction for multiple testing, throughout the entire study. Expectedly, this correction is more conservative than the FDR, so the number of CpG sites associated with candidate factors has now substantially decreased. Nevertheless, all our previous conclusions remain unchanged (e.g., enrichments for transcription factor binding sites, etc.).

6) Cell-type specific analyses: I understand that the authors use an interaction model, but I could not appreciate what is different in the mathematical exposition presented in this MS from the CellIDMC or TOAST methods published by other groups. Are the authors justified to include all these mathematical equations if they are not actually novel?

The reviewer is right in that we initially used interaction models similar to CellIDMC in order to identify cell type-specific effects on DNAm. However, due to the issues with interpretation and power mentioned by the reviewer (see comment below), we have changed this analysis and now

use, instead of interactions with cell proportions, interactions with contrasts between two pairs of cell proportions, defined by log-transformed ratios between proportions of the two cell subsets, as recommended in the literature on compositional data analysis. We have therefore removed the derivation of the previous interaction model, which is no longer used. This has now been clarified in the new version of the manuscript, in line 216 and in the Methods.

A related point is that when discussing the results of the cell-type specific analyses, they always report them “relative” to the neutrophil fraction, which I find rather confusing. Given that the neutrophil fraction is highest in blood, it is extremely important if the authors could report the number of cell-type specific changes in the neutrophil component in addition to the other cell-types. It should be noted that even with 1000 samples the authors could be seriously underpowered to detect changes in as many as 6 cell-types. One should also be careful in noting that the power to detect changes in a particular cell-type will depend on the variation in the fraction displayed by that cell-type, so this is variable between cell-types, which can make these type of analyses quite problematic. In view of this, the notion of cell-type specific needs to be clarified, since a reader may interpret this as changing in one of the 6 cell-type but not in the other 5, whilst someone else might interpret this as a statement of certainty that the change is happening in a given cell-type without necessarily implying how it changes in the other cell-types if at all. More clarification by the authors is required.

We agree with the reviewer that power is reduced when we estimate effects in several cell-types. We also understand that reporting results relative to the neutrophil fraction can be confusing. For these reasons, we have decided to estimate cell-specific effects of age, CMV infection, smoking and genotypes on DNAm in two pairs of blood cell compartments only: in lymphoid vs. myeloid lineages, as in You et al., *Nat Commun* 2020, or in naïve vs. memory T cells. Furthermore, we now use log-transformed ratios between proportions of the two cell subsets, as recommended in the literature on compositional data analysis. We found few cell-specific effects of age and CMV, suggesting that most of their effects on DNAm are common to blood cell types. In contrast, we found a substantial number of strong genotype × cell type effects ($n = 158$ CpG sites), suggesting that cell-specific mQTLs may be common.

These results are now reported in lines 215-223, 321-332 and 374-398.

Reviewer #2 (Remarks to the Author):

Bergstedt and colleagues report on a comprehensive analysis of factors associated with DNA methylation differences in blood. The cohort is beautiful with a large age range, equal numbers of men and women, a wealth of molecular and epidemiological data, and, crucially, detailed information on cell composition of the blood samples and CMV status is present. Also, the paper is well-written, and analyses generally seem to be well-thought through. The work has substantial potential to be of great interest to the population epigenetics community. While I started out reading the manuscript with great expectation, I was somewhat disappointed when I finished it. On the one hand I read many things I already knew from previous work (while science is about novelty), on the other hand I missed more depth regarding the analysis of aspects that are novel (cell counts and CMV effects, relative contribution of factors). A shift in focus to novelty would improve the paper, in particular when novel aspects were analyzed in detail than done now in a manuscript that deals with almost any aspect related to the field.

We thank the reviewer for their thoughtful comments. We are happy that the reviewer appreciates the usefulness of our cohort and the completeness of our analyses. We would like to clarify that, in the initial version of the manuscript, our objective was not to present known findings as if they were new, but to confirm several previous findings when correcting for cellular heterogeneity more completely than previous studies. Furthermore, one of our main objectives was to provide a quantitative assessment of the respective effects of the various factors known to affect DNAm, in the same study and cohort, so that they can be directly compared. Nevertheless, we understand that this can be perceived as less interesting than the newly described effects of CMV on DNAm variation and the comparison of corrections for cellular heterogeneity. Consequently, we have now extensively modified our manuscript, in order to give more emphasis on the most novel findings, in line with the reviewer's comments. Namely, we now highlight our results on (i) the effects of CMV infection on DNAm variation, including new analyses to evaluate its role as a possible mediator in EWAS and a new algorithm to predict CMV serostatus from DNA methylation data, and (ii) the comparison of corrections on cellular heterogeneity based on measured or estimated blood cell subsets.

We would like to note that, when revising the manuscript, we realized that our results were more robust when excluding 74 donors for which a large fraction of the flow cytometric data was missing. These missing values were initially imputed, but because imputation was not totally accurate, it introduced noise in the statistical inferences. All analyses are now done on 884 donors who have both accurate DNAm and flow cytometric data.

All revisions are highlighted *in gray* in the revised version of the manuscript.

- Repeatedly analysis and results are not explicitly described as a confirmation of previous work, work that often was more in depth than in the current manuscript that covers so many aspects. Readers may be left with the impression that the authors think results are new. When confirming previous results, a direct comparison of results is required (to what extent were the same CpGs found?) and a description of the new findings in addition to the previous work. Obviously, it should be made specific which previous work did the novel observation. That said: it is important for scientific progress to confirm previous findings. A non-complete list of examples includes:

In addition to the major changes described above, we have now verified that our results confirm previous findings and have cited systematically relevant studies, including Slieker et al Genome Biol 2016 (see lines 257-259, 288-290 and 339-341).

+ The manuscript presents as main finding (start discussion) a role of ZNF transcription factors, a role of KRAB, and SENP7 as example. SENP7 was originally found by Lemire et al (Nat Commun 2015). The result regarding ZNF transcription factors was a discovery reported by Hop et al (Genome Biol 2020). The discussion of the current manuscript reads as virtually identical to the latter paper (KRAB, deSUMOylation of KAP1). Both papers were already cited by the authors.

+ The characterization of meQTLs contains little news. There now are so many meQTL papers (including recently a meta-analysis of 30k samples; Min et al Nat Genet 2021). It remains unclear what new observations the authors made if any.

We have now moved most of the findings relating to meQTLs to supplementary text. We now only briefly mention them in the main text, in order to introduce the results relating to genotype-by-environment interactions. In addition, we have now clarified that our results only confirm previously published findings (see lines 360-373).

+ Epigenetic dispersion has been studied at a genome-wide level in detail previously. The authors may have missed the work of Zhang et al (Genome Med 2018) and Slieker et al (Genome Biol 2016). It will be informative to compare the CpGs common between the EPIC and 450k array. What is different? What insights are new? The interpretation of findings is very similar to the earlier reports, which is reassuring (and should be made explicit).

We thank the reviewer for this thoughtful comment. We have now compared the CpG sites that are the most dispersed with age with those found by a previous study (Slieker et al Genome Biol 2016). Reassuringly, we replicated 2,604 out of 5,075 CpG sites previously reported to be increasingly variable with age, supporting a strong overlap between the two different approaches (OR 95% CI: [36.2, 40.8]). Importantly, our model is expected to account for blood cell composition more completely than previous models, as it includes measured levels of memory T cells, which show a strong increase in variance with age (see new Fig. 3E). Furthermore, our model tests whether DNAm *changes* with age, which enabled us to determine if DNAm variance typically *increases* or *decreases* with age, contrarily to previous approaches. Strikingly, we found that 99.4% of CpGs with age-related dispersion show an increase in the variance of 5mC levels with age, in line with an age-related decrease in the fidelity of the epigenetic maintenance machinery. These results are further described in lines 280-303.

+ I enjoyed reading details like on DYRK2 promoter methylation in relation to sex-specific changes. It would be important to know if this is the first time that these pieces of information are linked, or this was already done before.

We have now clarified that this result is novel (line 344).

- Various aspects of the statistical analysis require more detail. Models fitted are complex with many (correlated) variables entered together, which requires reporting diagnostics to be sure nothing went wrong in fitting the models.

We agree with the reviewer that more details about the statistical analyses were required. We now show the Q-Q plot of residuals of models for the nine CpG sites most associated with age, sex and CMV serostatus, indicating that their error terms are normally distributed (see new Supplementary Fig. 5). We also plot the residuals of the models against age, sex, and CMV serostatus and show that the residuals are independent of these factors, which indicates that key model assumptions are fulfilled.

Also, comparisons of analysis with and without cell counts are only done at the level of genome-wide significant p-values, while a more explicit testing of hypotheses would lead to stronger inference.

We would like to clarify that our mediation analysis (see Equations 5 and 6) tests specifically if the effect of a candidate factor on DNAm levels at a CpG site is different when adjusting or not for blood cell composition. This has now been clarified further in the main text (lines 145-147).

+ Cell counts add up to 100%. This implies that models may not fit well when all counts are entered (e.g. issues with model intercept SE). Also, cell counts can be (strongly) correlated leading to collinearity. The authors should provide a heat map of correlations between their measured cell counts (perhaps combined with IDOL-predicted cell counts). Also, they need to address co-linearity and the fact that including all counts resulting in at least one parameter not being estimable.

We entirely agree with the reviewer that one parameter of cell proportions was not estimable in our previous analyses, which is why we did not interpret the effects of immune cell proportions on DNAm levels. Nevertheless, following the reviewer's comment, all the models where we adjust for blood cell composition (e.g., EWAS, meQTL mapping, etc.) are now fit without the proportion of neutrophils. Importantly, results based on these new models are similar to our previous results and all our previous conclusions remain unchanged. Of note, we would like to point out that, when the cell proportions are only covariates to adjust for, collinearity should not affect the interpretation of the parameter of interest (e.g., parameter for candidate factors such as age, CMV serostatus, genotypes...), except that it might lead to inflated standard errors.

Conversely, when estimating the effect on DNAm levels of *the immune cell proportions themselves*, collinearity can indeed be a problem. For this reason, in our new analyses, where we test for association between blood cell composition and DNAm levels directly (see new section lines 120-134), we now parametrize blood cell composition using mutually uncorrelated predictors, referred to as "contrasts". These predictors are constructed from a sequential binary partition, using tools from the literature in compositional data analysis (see the new Methods section for details).

Finally, as requested by the reviewer, we now provide a heat map of correlations between measured cell counts, as well as a heat map of correlations between *measured* and *estimated* cell proportions that are in common (see new Supplementary Fig. 3).

+ The authors need to report (and if necessary correct for) inflation of test statistics. In particular in combination with FDR, this may have large effects.

We thank the reviewer for this suggestion. We now report BACON inflation factors for all our EWAS (see new Supplementary Table 1).

+ The authors report the number of FDR-significant CpGs in a model without cell counts, with cell counts and those that are no longer FDR<0.05 with cell counts in the model. This provides a superficial and incomplete picture because very-low-p-value criteria are highly influenced by power instead of biology (slight change may render effect no longer significant).

To tackle this issue, we now report the factors that are the most associated with principal components of the DNAm data, adjusting or not on blood cell proportions. These results are now reported in lines 149-183.

We would also like to point out that we now use the Bonferroni correction for multiple testing throughout the entire study, as also requested by Reviewer 1. Expectedly, this correction is more conservative so the number of CpG sites associated with candidate factors has substantially decreased, but all our previous conclusions remain unchanged.

+ By comparing FDR significant findings from separate analyses, the authors cannot distinguish two effects of including cell counts: a mediator and noise-reduction (leading to more power).

+ While, the authors interpret their findings as mediation, this is not tested. The authors should explicitly perform mediation tests before they claim mediation (e.g. for CpGs with a ‘total effect’ only). Addition of mediation analysis is key in my opinion when improving the work.

It is important to clarify here that if key necessary assumptions hold, the models we fit to estimate mediated effects (Equations 3 and 4) are able to estimate and test for causal mediation, as described in Equation 5.1 in T.J. VanderWeele, in “Explanation in Causal Inference: Methods for Mediation and Interaction” (2015). To illustrate further that our approach tests for causal mediation, we have compared our results with those of the CMAverse R package (<https://bs1125.github.io/CMAverse/>), which is another implementation of the causal mediation analysis described in T.J. VanderWeele’s book (it was not available when we originally implemented our version). Using CMAverse, we estimated the mediation by immune cell proportions of the effect of age on DNAm levels, at the 10 CpG sites where we found the largest mediation effect in our original analysis. As expected, effect sizes and standard errors are identical, apart from bootstrap sampling errors. Estimates are shown in the Table A below. We apologize if we were not clear enough, and we hope that the reviewer now agrees that our initial models enable to formally test mediation.

CpG site	Our implementation		CMAverse	
	Estimate	SD	Estimate	SD
cg06419846	0.74	0.06	0.74	0.05
cg01877366	0.63	0.05	0.63	0.05
cg22213242	0.35	0.03	0.35	0.03
cg00151680	-0.63	0.05	-0.63	0.05
cg11569342	0.35	0.03	0.35	0.03
cg00259097	0.35	0.03	0.35	0.03
cg04249416	0.46	0.04	0.46	0.04
cg20793144	0.47	0.04	0.47	0.04
ch.13.39564907R	-0.6	0.05	-0.6	0.05
cg12484135	-0.6	0.05	-0.6	0.05

Table A. Comparison of mediation effect sizes computed with our model and with CMAverse

+ The cell types measured are of exceptional interest going from naïve to terminally differentiated cells. I was very eager to read more on the contribution of the various cell types. Simply entering all cell types provides little insight in the actual drivers of DNA methylation variation. We already know that cell counts matter a lot. The new data this study may bring is which specific cell types are important. More focus on this question (in combination with mediation analysis) would be of great interest to the field.

To address the reviewer's points, we have now included a new Results section relating to the effects of blood cell composition on DNAm, where we compare EWAS results corrected for 16 cell types measured by FACS or cell types estimated by Houseman *et al.*'s method, IDOL method or the recently published EPIC IDOL-Ext method. These results suggest that the adjustment for proportions of CD4⁺ and CD8⁺ memory T cells is essential to account more completely for blood cell composition. Accordingly, we found that, in a model including all the measured cell types, as well as age, sex, smoking and CMV, the ratio of myeloid vs. lymphoid cell lineages and the ratio of naïve vs. memory CD4⁺ and CD8⁺ T cells are the most strongly associated with PC1 of DNAm. These results are now reported in lines 120-134 and 168-193.

Regarding cell type-specific effects on DNAm, we entirely agree with the reviewer that such analyses are of great interest, which is why we sought to estimate, in the initial version of the manuscript, cell type-specific effects of age and meQTLs. To address the reviewer's comment, we have now added an analysis of the direct impact of blood cell composition on DNAm levels. We show that the ratio between myeloid and lymphoid cells is associated with 24% of all CpG sites. Strikingly, the ratio between naïve and memory CD8⁺ T cells is associated with 7% of all CpG sites, suggesting that differences in the proportion of naïve and differentiated subsets of CD4⁺ and CD8⁺ T cells contribute substantially to DNA methylation variation and may mediate associations between DNA methylation and environmental exposures or diseases.

Furthermore, we have now extended our previous interaction analyses to the cell-type-specific effects of CMV infection and smoking (see lines 215-223 and Supplementary Text). Importantly, because Reviewer 1 has raised important concerns regarding the power to detect such effects, particularly when the number of included cell-types is large, we have decided to estimate cell-specific effects of age, CMV infection, smoking and genotypes on DNAm for two pairs of blood cell compartments only: in lymphoid vs. myeloid lineages, as in You et al., *Nat Commun* 2020, or in naïve vs. memory T cells. We found few cell-specific effects of age and CMV, suggesting that most of their effects on DNAm are common to blood cell types. In contrast, we find a substantial number of strong genotype × cell type effects ($n = 158$ CpG sites), suggesting that cell-specific meQTLs are common. These results are now reported in lines 321-332 and 374-398.

+ *Similar reasoning is valid for CMV: the observation that CMV matters (and not other viruses) is highly relevant, but is overshadowed in the current manuscript by findings repeating previous work, while more in-depth analyses regarding CMV would be of great interest.*

Following the reviewer's suggestions, we have now added several new results that describe the effects of CMV on 5mC levels in human blood, including (i) EWAS that are adjusted or not on CMV; (ii) associations of CMV with principal components of the DNAm data; (iii) CMV-mediated effects of age; (iv) cell-type-specific effects of CMV (see our response to previous comment) and (v) a predictive model of CMV serostatus from DNAm data, to enable future EWAS to potential control for confounding by CMV.

Our new results show that, when not adjusting for cell proportions, or when adjusting on cells estimated by Houseman *et al.*'s or IDOL methods, CMV serostatus is strongly associated with PC1 and PC2, whereas age and sex explain PC3 to PC5. However, when adjusting on 16 measured cell proportions, PC1 and PC2 are no longer associated with CMV, which are actually explained by immune cell proportions, including CD4⁺ and CD8⁺ memory T cells. These results imply that most CMV effects on DNAm are mediated by blood cell composition, in line with our initial conclusions and the fact that CMV infection causes a strong increase in memory T cell subsets (Patin*, Hasan*, Bergstedt* et al., *Nat Immunol* 2018). Importantly, naïve and memory T cell fractions are not estimated by standard cell mixture deconvolution methods, which suggests that CMV can affect EWAS results when these methods are used.

We also found statistical evidence at 10,074 CpG sites that CMV mediates total effects of age on DNAm. Accordingly, age has a significant total effect on 5mC levels at 97,219 and 113,742 CpG sites when adjusting or not on CMV serostatus, which is why we revisited the well-described age effects on DNAm, but adjusting on CMV. Expectedly, we found that CMV infection does not mediate the effects of smoking, sex, BMI or CRP levels as these factors are not associated with CMV serostatus. We anticipate that CMV may confound EWAS of disease associated with CMV, such as cardiovascular disease (Wang et al., *J Am Heart Assoc* 2017).

All these new results are now included in the revised manuscript, lines 149-167 and 195-246.

- Is there evidence of dispersion in cell counts with age? Important to report.

We agree with the reviewer that this is an important point. We have now added a new plot showing that cell proportions, memory CD4⁺ T cells in particular, show increased dispersion with age (see new Fig. 3E). We believe that this further highlights the need to adjust age-dispersion models on blood cell composition measured (or estimated) at high resolution.

- Smoking-associated differential DNA methylation at GPR15 was suggested to be induced by T cell composition changes (Bauer et al. Clin Epigenet 2015). Is this something the authors could confirm or exclude (in relation to the conclusion that smoking-associated methylation differences are often not related to cell counts).

To confirm or exclude this intriguing finding, we verified if the effect of smoking on cg19859270 (i.e., the CpG site of interest in *GPR15*) is mediated or not by changes in blood cell composition. We found that smoking has a strong total and direct effect on cg19859270 ($P_{\text{adj}} = 5.8 \times 10^{-31}$ and $P_{\text{adj}} = 3.6 \times 10^{-29}$, respectively), but no cell-composition-mediated effect ($P_{\text{adj}} = 1.0$). In addition, we verified if cg19859270 is more strongly associated with smoking status in lymphoid vs. myeloid lineages, by using an interaction model. We found a suggestive association that is not significant before correction for multiple testing ($P = 0.077$). Overall, this suggests that, in our cohort, most of the effects of smoking on *GPR15* DNAm levels are similar across blood cell types. We note however that our results do not totally exclude the possibility that smoking induces the increase of CD3⁺ GPR15⁺ T cells, as these cells have not been measured in our cohort. These results are now described in the new Supplementary Text.

- Heparin negatively affects DNA studies. How did the authors address this problem?

We thank the reviewer for this comment, which made us realize that the initial version of the manuscript was unclear regarding this aspect. We have now clarified that different anticoagulants were used, depending on the downstream analyses: for DNA methylation profiling, blood samples were collected on EDTA, whereas samples for flow cytometry and genome-wide DNA genotyping were collected on Li-heparin (see lines 538-540). Furthermore, we would like to note that two studies have showed that the treatment of fresh blood samples by different anticoagulants (EDTA, heparin) have no detectable effects on DNA methylation levels (Hebels et al., *Environ Health Perspect* 2013; Shiwa et al., *PLoS One* 2016).

- The authors excluded 118k probes overlapping with any SNP MAF>1%. Was this irrespective of the exact location of the SNP relative to the probe and, if so, why?

We excluded any probe overlapping a SNP within the 50 pb of the CpG site, to ensure that no meQTL signals are due to allele-dependent DNA binding. This is now clarified in line 581.

- Data are not yet available from EGA? The authors are to be commended for creating a browser that enables others to download summary stats.

The data will be made available upon manuscript publication, through our new institutional data repository, OWEY, recently developed by Institut Pasteur, which will avoid previous challenges and delays that we have experienced with EGA.

- Why did the authors not for example use PBMC chromatin states? Choice for a single cell type is unclear.

We now report results according to the ENCODE chromatin states measured in PBMCs. All our previous conclusions remain unchanged.

REVIEWER COMMENTS

Reviewer #1 (Remarks to the Author):

The revised version of the paper "Factors Driving DNA Methylation Variation in Human Blood" is a significant improvement. The paper now delves straight into the interesting and novel stuff, and contains some very important findings for the whole epigenome community. Notably, the need to adjust for cell-type heterogeneity at a higher cellular resolution is well demonstrated, as well as the effect of CMV infection on cell-type composition and DNAm variation.

However, I still have some major technical concerns in relation to the cell-type specific analyses, which need to be addressed, as I am not at all convinced that the statistical methods being implemented in these analyses are correct or well-justified. In addition, I would also question that some of the interpretations derived from these cell-type specific analyses are correct, as indeed some of the results contradict recent current literature:

1) Fig.4E (left panel): would it not be better to display the left panel with the x-axis labeling myeloid proportion? After all, the authors claim the effect is in dendritic cells which is part of the myeloid compartment, and so the difference in DNAm between genotypes should increase with increased myeloid fraction. Follow-up questions: is there any visual evidence of an interaction if they plot DNAm vs the dendritic fraction? And is it just dendritic cells that matter here or how about monocytes (which I understand make up the larger cell-type proportion in blood?).

2) Example in Fig.4E is not truly myeloid specific: An important follow-up point to the above is that if we extrapolate the fitted regression lines in left panel of Fig.4E to when the lymph fraction is close to 1 (this I know is unrealistic but statistically this is how we can establish if the mQTL is truly myeloid-specific, meaning it is not present in lymphocytes), we can see that even in very high lymph fraction samples, there is a difference in DNAm between the genotypes. In other words, what the authors call "myeloid-specific" and the example provided in Fig.4E is not strictly speaking "myeloid-specific" because the mQTL effect is still present in lymphocytes, albeit less strongly so. So, would it not be safer to state that the mQTL example in Fig.4E is predominantly myeloid?

3) Compositional analysis (lines 627-641): there are a number of important issues relating to this paragraph in Methods section which I am somewhat confused about. First of all, the first 3 sentences of this paragraph seem contradictory to me. In the first two sentences, the authors state that because fractions must be non-negative and add to 1 that these fractions should not be estimated with unconstrained methods. In the subsequent sentence the authors state that they use a method to transform fraction coordinates into a basis that is free of any constraints, i.e. unconstrained. So, I am confused by these two contradictory statements. Second, the authors cite two rather old papers from the statistical literature, and don't provide any validation of these methods on real DNAm datasets. Third, I would disagree with the 2nd sentence in this paragraph. There is substantial evidence from the literature that unconstrained methods work *better* than the constrained projection (CP) method introduced by Houseman. I can refer the authors to work from Newman A et al CIBERSORT Nat Methods, as well as to the paper by Teschendorff BMC Bioinformatics 2012 (cited in this MS), where it is shown that imposing the constraints only after inference of the regression weights (hence unconstrained) may work better than CP. Fourth, how exactly the estimated fractions are made to add-up to 1 (if at all) is unclear, which raises the follow-up question as to how exactly you can interpret the actual estimates, which the authors call "contrasts". Fifth, can the authors explain why they end up with only 14 contrasts, and not 15? After all, they have 16 cell-type fractions to estimate, and only 15 are independent. So why 14? Or assume we only have 2 cell-types, then we should have $2-1=1$ contrast, so why for 16 the authors get 14 and not 15? Sixth, as described in this paragraph, the approach to estimate all 14 contrasts is hierarchical. I like this concept a lot and indeed a previous method called HEpiDISH (see Zheng SC et al Epigenomics 2018) also tries out such a hierarchical approach. Yet exactly why these contrasts are "hierarchical" is unclear since the authors do not provide any details or equations. Nor do the authors validate the estimated contrasts using say simulation models. Seven, if the authors are using a novel method that has not been validated or

extensively validated on DNAm datasets, then it is imperative that the authors provide FULL details of validations in this paper (once again using simulation models). Eight, it is terribly unclear to me in which analyses these particular 14 contrasts were used, or why they are used instead of measured FACS-based values? For instance, from the next subsections in Methods I get the impression that these contrasts were used when correlating PCs to cell-type fractions, or when doing the association analyses between DNAm and cell-type composition (e.g. Eqn-2 on line 661), but from the Results section this is not clear...

4) Cell-type specific models (Eqns-9 & 10, lines 812-818) are not justified: I am sorry, but I find the description of these cell-type specific analyses very unclear and vague, and indeed I could not find any statistical justification for why the authors use the logarithm of the ratio of two contrast values? Moreover, the authors seemingly perform one analysis at the myeloid-lymphoid resolution (this I can understand) but then another to distinguish naïve from mature T-cell compartments. However, Eqn.11 is plainly wrong if you just substitute the $\log(c_{naive}/(1-c_{naive}))$ for c_i , because this ignores the variation in DNAm associated with all other cell-types, notably the myeloid cells. In other words, the DNAm values appearing on the LHS of the equation are not renormalized to subtract the contribution from myeloid cells. Again, there is absolutely no justification or validation for the statistical models being implemented: to me appear these models appear as being completely ad-hoc.

5) Cell-type specific mQTLs: adding to my concern above, the authors also make a rather surprising conclusion which contradicts much of the current and recent literature on cell-type specific mQTLs. For instance, the recent mQTL paper by John Chambers, see *Hawe JS Nat Genet 2022*, clearly indicates that the great majority of mQTLs are not cell-type specific, even when comparing across disparate cell-types like fat and immune cells. This result is consistent with findings from BLUEPRINT (see Chen, Soranzo N et al *Cell 2016*) where they also estimated that over 90% of mQTLs are independent of blood cell subtype. The paper You C et al *Nat Commun 2020* also includes power calculations suggesting that it might be difficult to detect non cell-type specific mQTLs in both lymphoid and myeloid compartments, leading to the illusion that many are myeloid specific. Therefore, I am a little worried that the statements on lines 375-376 being incorrect or misleading, which may also have to do with the statistical models (Eqns-9-11) not being statistically well motivated. I strongly advise the authors to fully explain and justify their models, include some validation on simulated data, or alternatively to substantially cut or tone down the cell-type specific analyses, as some of the associated findings could be wrong. I note that if an mQTL is not cell-type specific, that this still allows for the possibility that the effect on gene-expression is cell-type specific.

6) Clarity of mathematical equations could be improved: related to the above points, I would also encourage the authors to adopt a clearer notation when writing down their mathematical equations. For instance, indices like "p" or "k" appear on the LHS but not on the RHS. What is a vector and what is not would also be unclear to the wider audience.

In summary, I think that this is an excellent paper, but the statistical models being used for some analyses, specially the compositional and cell-type specific ones, appear completely ad-hoc and have not been properly tested. I advise the authors to either cut some of these out, or to use more traditional methods (ie fractions as opposed to contrasts), or alternatively to provide full details and validations of these models.

Reviewer #2 (Remarks to the Author):

I appreciated the careful additional analyses and much enjoyed reading the revised version. I am convinced the work will be of considerable interest to the field.

Some minor thoughts that came up – I leave it at the discretion of the authors whether or not to address them.

- Gene name: EVOLV2 > ELOVL2(?)

- The authors refer to 'Cell and tissue type independent age-associated DNA methylation changes are not rare but common' (47), in which a study claiming the opposite is discussed. Perhaps, the current in-depth analysis can provide some balance in the debate (as far as blood cells are concerned).

- The insightful analysis as to the role of various cell types to age-related changes reminded me of a recent study indicating that epigenetic clocks work by marking age-related changes in naïve/activated T cell subsets, which seems in line with the current data (Jonkman et al. Genome Biol 2022).

REVIEWER COMMENTS

Reviewer #1 (Remarks to the Author):

The revised version of the paper “Factors Driving DNA Methylation Variation in Human Blood” is a significant improvement. The paper now delves straight into the interesting and novel stuff, and contains some very important findings for the whole epigenome community. Notably, the need to adjust for cell-type heterogeneity at a higher cellular resolution is well demonstrated, as well as the effect of CMV infection on cell-type composition and DNAm variation.

However, I still have some major technical concerns in relation to the cell-type specific analyses, which need to be addressed, as I am not at all convinced that the statistical methods being implemented in these analyses are correct or well-justified. In addition, I would also question that some of the interpretations derived from these cell-type specific analyses are correct, as indeed some of the results contradict recent current literature: In summary, I think that this is an excellent paper, but the statistical models being used for some analyses, specially the compositional and cell-type specific ones, appear completely ad-hoc and have not been properly tested. I advise the authors to either cut some of these out, or to use more traditional methods (ie fractions as opposed to contrasts), or alternatively to provide full details and validations of these models.

RESPONSE: We warmly thank the reviewer for their precise, insightful comments. We are pleased to know that they find our paper excellent and that it includes important findings for the whole epigenome community. To address the reviewer’s concerns regarding our compositional and cell-type dependent analyses, we have now:

- validated our compositional analyses both theoretically (with simulations) and empirically (comparing our results with DNAm data in sorted cells),
- conducted cell-type-dependent analyses using cell proportions rather than contrasts,
- clarified extensively the Methods sections.
- clarified that we have not searched for cell-type-specific effects but cell-type-dependent effects.

The detailed changes introduced in the new revised version of the manuscript are described below.

1) Fig.4E (left panel): would it not be better to display the left panel with the x-axis labeling myeloid proportion? After all, the authors claim the effect is in dendritic cells which is part of the myeloid compartment, and so the difference in DNAm between genotypes should increase with increased myeloid fraction. Follow-up questions: is there any visual evidence of an interaction if they plot DNAm vs the dendritic fraction? And is it just dendritic cells that matter here or how about monocytes (which I understand make up the larger cell-type proportion in blood?).

RESPONSE: Following the reviewer’s suggestion, we now show *CLEC4C* DNAm levels as a function of the proportion of myeloid cells, in the new Fig. 4E. Furthermore, we tested if the genotype × cell-type proportion interaction observed at *CLEC4C* for the myeloid fraction is also detected for dendritic cells and monocytes (Fig. A below, for the reviewer). We found a strongly significant interaction for dendritic cells (interaction effect, 95% CI: CI: [-8.3, -5], $P_{\text{adj}} = 3.7 \times 10^{-12}$) and a non-significant interaction for monocytes (interaction effect, 95% CI: [-0.42, 0.002], $P_{\text{adj}} = 0.07$), in agreement with a DC-dependent effect of rs11055602 on *CLEC4C* DNAm levels.

These results are now described in the revised manuscript (l. 388-390).

Figure A. Cell-type-dependent effects of the rs11055602 variant on 5mC levels at the *CLEC4C* locus, for dendritic cells (left) and monocytes (right).

2) Example in Fig.4E is not truly myeloid specific: An important follow-up point to the above is that if we extrapolate the fitted regression lines in left panel of Fig.4E to when the lymph fraction is close to 1 (this I know is unrealistic but statistically this is how we can establish if the mQTL is truly myeloid-specific, meaning it is not present in lymphocytes), we can see that even in very high lymph fraction samples, there is a difference in DNAm between the genotypes. In other words, what the authors call “myeloid-specific” and the example provided in Fig.4E is not strictly speaking “myeloid-specific” because the mQTL effect is still present in lymphocytes, albeit less strongly so. So, would it not be safer to state that the mQTL example in Fig.4E is predominantly myeloid?

RESPONSE: We fully agree with the reviewer and apologize for the inappropriate use of the term “cell-type-specific”. **We have now replaced “cell-type-specific” by “cell-type-dependent”**, throughout the entire manuscript.

3) Compositional analysis (lines 627-641): there are a number of important issues relating to this paragraph in Methods section which I am somewhat confused about. First of all, the first 3 sentences of this paragraph seem contradictory to me. In the first two sentences, the authors state that because fractions must be non-negative and add to 1 that these fractions should not be estimated with unconstrained methods. In the subsequent sentence the authors state that they use a method to transform fraction coordinates into a basis that is free of any constraints, i.e. unconstrained. So, I am confused by these two contradictory statements.

I would disagree with the 2nd sentence in this paragraph. There is substantial evidence from the literature that unconstrained methods work **better** than the constrained projection (CP) method introduced by Houseman. I can refer the authors to work from Newman A et al CIBERSORT Nat Methods, as well as to the paper by Teschendorff BMC Bioinformatics 2012 (cited in this MS), where it is shown that imposing the constraints only after inference of the regression weights (hence unconstrained) may work better than CP.

RESPONSE: We agree that this section was unclear. Following the reviewer’s advice, **we have now clarified this entire section (l. 642-679)**. Namely, we now describe more clearly in this section that our

aim is to estimate the association between cell composition, measured by flow cytometry, and DNA methylation. To do so, we would need to fit a linear regression model, for each CpG site, with 5mC levels as response variable and 16 measured proportions of immune cells as predictors (see new *Equation 3*). However, the fact that the 16 immune cell proportions are constrained to be positive and sum to one makes the interpretation of effect sizes of such models difficult (see for instance, Dumuid et al. *Stat Methods Med Res* 2018; Arnold et al. *Int J Epidemiol* 2020). To get more interpretable estimates, we employ methods from compositional data analysis (see, for instance, Pawlowsky-Glahn, Egozcue, Tolosana-Delgado 2015). Namely, we transform the measured proportions into unconstrained variables (i.e., log-ratios, or balances, see new *Equations 1* and *2* for examples) and use them as predictors in linear regression models. We would like to clarify that we did not mean that constrained methods are more accurate than unconstrained methods and apologize for the confusion created.

Second, the authors cite two rather old papers from the statistical literature, and don't provide any validation of these methods on real DNAm datasets.

RESPONSE: To address the reviewer's comment, we have now validated our compositional analyses both theoretically and empirically. First, we have used simulations of cellular fractions assuming a Dirichlet distribution, to evaluate the accuracy of our compositional analyses. Reassuringly, we find that our model estimates the effects that are expected according to the simulations (new Supplementary Fig. 2). Second, we have retrieved publicly available EPIC array data for 12 sorted immune cell subsets (Salas et al., *Nat Commun* 2022) and compared our estimated effects of immune cell balances on DNAm levels to the mean DNAm differences between sorted cell subsets. We find that effects of the myeloid/lymphoid balance (see new *Equation 1*) on DNAm are strongly correlated with DNAm differences between myeloid and lymphoid lineages (Pearson's $R = 0.92$), while being much less correlated with differences between other pairs of immune cell subsets (**new Supplementary Table 2**). We observed similar results for other balances ($R > 0.6$). Collectively, these new results suggest that our compositional analyses and their interpretation are accurate.

These new results are now included in l. 124-128, new Supplementary Fig. 2, Supplementary Text and Supplementary Table 2.

Fourth, how exactly the estimated fractions are made to add-up to 1 (if at all) is unclear, which raises the follow-up question as to how exactly you can interpret the actual estimates, which the authors call "contrasts".

RESPONSE: We would like to clarify that cell proportions were not estimated, but experimentally measured by flow cytometry, using a hierarchical gating strategy (Patin*, Bergstedt*, Hasan* et al., *Nat Immunol* 2018). Consequently, measured cell proportions are already expected to sum to one. Yet, because of measurement errors, cell fractions do not exactly sum to one in all donors, reason by which we used, as a measure of the proportion of given cell subset in a given donor, the absolute count of this cell-type divided by the sum of absolute counts of all the 16 measured cell subsets.

This has been now clarified in the Methods (l. 634-638).

Regarding the interpretation of contrasts (or balances), they are computed from a transformation of immune cell proportions that has been developed in the field of compositional data analysis. The transformation is the logarithm of the ratio between proportions of two groups of cell-types and can be seen as generalization of the logit-transformation (see new *Equations 1* and *2* for examples). The effect of the balance between lymphoid and myeloid cells on DNA methylation is interpreted as the change in 5mC levels for an increase in myeloid cells and the corresponding decrease in lymphoid cells.

This has also been clarified in the Methods (l. 642-679).

Fifth, can the authors explain why they end up with only 14 contrasts, and not 15? After all, they have 16 cell-type fractions to estimate, and only 15 are independent. So why 14? Or assume we only have 2 cell-types, then we should have $2-1=1$ contrast, so why for 16 the authors get 14 and not 15?

RESPONSE: We fully agree with the reviewer and apologize for the confusion. In the previous version of the manuscript, we excluded dendritic cells from the contrasts, because they represent < 0.5% of circulating blood cells. Therefore, we thought that the effects of this cellular fraction on DNAm variation would be small. We now report results for the 15th missing balance, the eosinophil / DC balance, which is associated with 5,010 CpG sites.

These results are now reported in the new Supplementary Table 2.

Sixth, as described in this paragraph, the approach to estimate all 14 contrasts is hierarchical. I like this concept a lot and indeed a previous method called HEpiDISH (see Zheng SC et al Epigenomics 2018) also tries out such a hierarchical approach. Yet exactly why these contrasts are “hierarchical” is unclear since the authors do not provide any details or equations. Nor do the authors validate the estimated contrasts using say simulation models. Seven, if the authors are using a novel method that has not been validated or extensively validated on DNAm datasets, then it is imperative that the authors provide FULL details of validations in this paper (once again using simulation models).

RESPONSE: As mentioned above, we have now validated our compositional analyses both theoretically and empirically. We believe that these new results provide strong support in favor of our hierarchical approach. Furthermore, we have added a clearer description of this procedure in the Methods (l. 642-679). Please note that our sequential binary partition, which defines the immune cell hierarchies we used, is **reported in the new Supplementary Table 2.**

Eight, it is terribly unclear to me in which analyses these particular 14 contrasts were used, or why they are used instead of measured FACS-based values? For instance, from the next subsections in Methods I get the impression that these contrasts were used when correlating PCs to cell-type fractions, or when doing the association analyses between DNAm and cell-type composition (e.g. Eqn-2 on line 661), but from the Results section this is not clear...

RESPONSE: Again, we are sorry for this confusion. We have now clarified throughout the entire manuscript that balances were used only when estimating the effects of cell composition on 5mC levels, measured at either individual CpG sites or summarized by principal components (see l. 121-141, l. 676-679).

4) Cell-type specific models (Eqns-9 & 10, lines 812-818) are not justified: I am sorry, but I find the description of these cell-type specific analyses very unclear and vague, and indeed I could not find any statistical justification for why the authors use the logarithm of the ratio of two contrast values?

RESPONSE: We agree that the models we used are not standard (i.e., models with a factor \times cell-type balance interaction term). Following the reviewer’s advice, we now use the proportion of lymphoid cells as the interacting variable for all cell-type dependent analyses.

The cell-type dependent analyses have now been updated (l. 224-227, l. 327-331, l. 381-390) and the corresponding section of the Methods has been changed accordingly (l. 873-891).

Moreover, the authors seemingly perform one analysis at the myeloid-lymphoid resolution (this I can understand) but then another to distinguish naïve from mature T-cell compartments. However, Eqn.11 is plainly wrong if you just substitute the $\log(c_{naive}/(1-c_{naive}))$ for c_i , because this ignores the variation in DNAm associated with all other cell-types, notably the myeloid cells. In other words, the DNAm values appearing on the LHS of the equation are not renormalized to subtract the contribution from myeloid

cells. Again, there is absolutely no justification or validation for the statistical models being implemented: to me appear these models appear as being completely ad-hoc.

RESPONSE: To address the reviewer's comments, we have now removed from the entire manuscript the cell-type-dependent analyses for naïve vs differentiated T cells.

5) *Cell-type specific mQTLs: adding to my concern above, the authors also make a rather surprising conclusion which contradicts much of the current and recent literature on cell-type specific mQTLs. For instance, the recent mQTL paper by John Chambers, see Hawe JS Nat Genet 2022, clearly indicates that the great majority of mQTLs are not cell-type specific, even when comparing across disparate cell-types like fat and immune cells. This result is consistent with findings from BLUEPRINT (see Chen, Soranzo N et al Cell 2016) where they also estimated that over 90% of mQTLs are independent of blood cell subtype. The paper You C et al Nat Commun 2020 also includes power calculations suggesting that it might be difficult to detect non cell-type specific mQTLs in both lymphoid and myeloid compartments, leading to the illusion that many are myeloid specific. Therefore, I am a little worried that the statements on lines 375-376 being incorrect or misleading, which may also have to do with the statistical models (Eqns-9-11) not being statistically well motivated.*

RESPONSE: We would like to clarify that, out of 107,048 CpG sites that show a significant local meQTL, only 249 CpG sites showed a cell-type-dependent genetic effect, thus representing 0.23% of CpG sites. This figure is thus consistent with previous studies (e.g., Hawe et al., *Nat Genet* 2022). To avoid any confusion or ambiguity, **we have now made it clear in the revised manuscript (l. 383-384; l. 405-406).**

I strongly advise the authors to fully explain and justify their models, include some validation on simulated data, or alternatively to substantially cut or tone down the cell-type specific analyses, as some of the associated findings could be wrong. I note that if an mQTL is not cell-type specific, that this still allows for the possibility that the effect on gene-expression is cell-type specific.

RESPONSE: To detect cell-type-dependent meQTLs, we initially used a regression model with an interaction term between genotypes at a SNP and a cell balance. We agree that the use of cell balances in this context is uncommon and may require further justification. To address the reviewer's concern, we have now conducted these analyses with the myeloid proportion as the interacting term, which has been used regularly in the field (see for example, Hawe et al., *Nat Genet* 2022).

The new results, which are very similar to previous results, are now reported in the new manuscript (l. 381-390).

6) *Clarity of mathematical equations could be improved: related to the above points, I would also encourage the authors to adopt a clearer notation when writing down their mathematical equations. For instance, indices like "p" or "k" appear on the LHS but not on the RHS. What is a vector and what is not would also be unclear to the wider audience.*

RESPONSE: We thank the reviewer for this comment. **We have now revised all equations accordingly.**

Reviewer #2 (Remarks to the Author):

I appreciated the careful additional analyses and much enjoyed reading the revised version. I am convinced the work will be of considerable interest to the field.

RESPONSE: We warmly thank the reviewer for their insightful comments. We are pleased to know that they are convinced that our work will be of considerable interest to the field.

Some minor thoughts that came up – I leave it at the discretion of the authors whether or not to address them. Gene name: EVOLV2 > ELOVL2

RESPONSE: We thank the reviewer for pointing this out. The gene name has now been corrected (see **l. 265**).

- The authors refer to ‘Cell and tissue type independent age-associated DNA methylation changes are not rare but common’ (47), in which a study claiming the opposite is discussed. Perhaps, the current in-depth analysis can provide some balance in the debate (as far as blood cells are concerned).

RESPONSE: We have now further clarified the sentence. In good agreement with Zhu *et al.* (Aging 2018), we find little evidence for cell-type-dependent effects of age on DNA methylation: only 10 CpG sites are associated with age more strongly within myeloid cells than in lymphoid cells (see **l. 327-338**).

*- The insightful analysis as to the role of various cell types to age-related changes reminded me of a recent study indicating that epigenetic clocks work by marking age-related changes in naïve/activated T cell subsets, which seems in line with the current data (Jonkman *et al.* Genome Biol 2022).*

RESPONSE: We thank the reviewer for this comment. We agree that this article is of great relevance to our study. We now cite this reference when we show that proportions of naïve and differentiated T cells largely contribute to population variation in DNA methylation (see **l. 136-137**).

REVIEWERS' COMMENTS

Reviewer #1 (Remarks to the Author):

I am happy with the revisions made to this last version and I look forward to seeing this published.